# High-Resolution Proteomics Unravel a Native Functional Complex of Cav1.3, SK3, and Hyperpolarization-Activated Cyclic Nucleotide-Gated Channels in Midbrain Dopaminergic Neurons

**DOI:** 10.3390/cells13110944

**Published:** 2024-05-30

**Authors:** Maya Belghazi, Cécile Iborra, Ophélie Toutendji, Manon Lasserre, Dominique Debanne, Jean-Marc Goaillard, Béatrice Marquèze-Pouey

**Affiliations:** 1CRN2M Centre de Recherche Neurobiologie-Neurophysiologie, CNRS, UMR7286, Aix-Marseille Université, 13015 Marseille, France; maya.belghazi@univ-amu.fr; 2Institut de Microbiologie de la Méditerranée (IMM), CNRS, Aix-Marseille Université, 13009 Marseille, France; 3Ion Channel and Synaptic Neurobiology, INSERM, UMR1072, Aix-Marseille Université, 13015 Marseille, France; cecile.iborra@univ-amu.fr (C.I.); ophelie.toutendji@gmail.com (O.T.); manon.lasserre@laposte.net (M.L.); dominique.debanne@univ-amu.fr (D.D.); jean-marc.goaillard@univ-amu.fr (J.-M.G.); 4Institut de Neurosciences de la Timone, CNRS, Aix-Marseille Université, 13005 Marseille, France

**Keywords:** substantia nigra, ion channels, protein–protein interaction, affinity purification mass spectrometry, proximity ligation assay, pacemaking

## Abstract

Pacemaking activity in substantia nigra dopaminergic neurons is generated by the coordinated activity of a variety of distinct somatodendritic voltage- and calcium-gated ion channels. We investigated whether these functional interactions could arise from a common localization in macromolecular complexes where physical proximity would allow for efficient interaction and co-regulations. For that purpose, we immunopurified six ion channel proteins involved in substantia nigra neuron autonomous firing to identify their molecular interactions. The ion channels chosen as bait were Cav1.2, Cav1.3, HCN2, HCN4, Kv4.3, and SK3 channel proteins, and the methods chosen to determine interactions were co-immunoprecipitation analyzed through immunoblot and mass spectrometry as well as proximity ligation assay. A macromolecular complex composed of Cav1.3, HCN, and SK3 channels was unraveled. In addition, novel potential interactions between SK3 channels and sclerosis tuberous complex (Tsc) proteins, inhibitors of mTOR, and between HCN4 channels and the pro-degenerative protein Sarm1 were uncovered. In order to demonstrate the presence of these molecular interactions in situ, we used proximity ligation assay (PLA) imaging on midbrain slices containing the substantia nigra, and we could ascertain the presence of these protein complexes specifically in substantia nigra dopaminergic neurons. Based on the complementary functional role of the ion channels in the macromolecular complex identified, these results suggest that such tight interactions could partly underly the robustness of pacemaking in dopaminergic neurons.

## 1. Introduction

Midbrain substantia nigra pars compacta (SNc) dopaminergic (DA) neurons are involved in motor control and action selection. They display an autonomous tonic regular firing in the absence of synaptic inputs, which maintains the extracellular tone of dopamine necessary for the proper functioning of target structures, such as the striatum (see [1] for a review). Progressive loss of SNc DA neurons is responsible for the primary motor symptoms of Parkinson’s Disease (see [2] for a review), and several studies have suggested that their vulnerability might be linked to their peculiar electrophysiological properties, and, particularly, their pacemaking activity [3].

Numerous types of somatodendritic voltage- and calcium-gated ion channels act synergistically in this intrinsic rhythmicity, inducing sub-threshold depolarizations and oscillations, action potentials, and repolarization of the membrane potential following the action potential (see [1] for a review).

The goal of our molecular approach was to understand whether the functional cooperation between ion channels underlying pacemaking activity could arise from a common localization in macromolecular complexes where physical proximity would allow for efficient interactions and co-regulations. In order to test this hypothesis, we focused on several somatodendritic ion channels previously reported as particularly relevant to the rhythmicity of SNc DA neurons.

There is a general consensus that L-type voltage-dependent Ca^2+^ channels, including, in particular, Cav1.2 (*Cacna1c* gene) and Cav1.3 (*Cacna1d* gene) α1-subunits [4,5], are essential for the generation of pacemaking activity, even though the use of L-type channel antagonists has led to contrasting results [6,7,8,9]. Both L-type channel subunits are expressed in SNc neurons. These proteins share 75% coding sequence homology, but the currents they carry differ significantly, with Cav1.3 channels opening at a relatively hyperpolarized, sub-threshold membrane potential (∼−50 mV), approximately 25 mV more hyperpolarized than Cav1.2 [10]. Due to these differences in voltage sensitivity, Cav1.3 has been reported to have a much stronger role than Cav1.2 in pacemaking [7].

Each action potential, produced mainly through the activation of transient sodium channels, is followed by an apamin-sensitive afterhyperpolarization (AHP) [11] generated by calcium-activated small-conductance SK potassium channels. SK3 channels (also named K_Ca_2.3, *Kcnn3* gene) were found to be the main paralogous proteins enriched in SNc DA neurons [12,13], responsible for most of the AHP current that controls firing regularity [14]. Using computational, imaging, and electrophysiological approaches, Wilson and Callaway predicted a functional coupling between a low-threshold activated L-type Ca^2+^ (i.e., Cav1.3) and SK channels [15]. Moreover, low-voltage-activated T-type Ca^2+^ channels were also shown to be coupled to SK channels in SNc DA neurons [16].

Following the AHP, a slowly activating nonselective cation current (I_H_) mediated by the hyperpolarization-activated cyclic nucleotide-gated (HCN) channels plays a role in resuming pacemaker firing in most of the SNc DA neurons [17,18]. Co-expression of HCN2, HCN3, and HCN4 mRNAs coding for the slower I_H_ channels was detected in single SNc DA neurons [19]. HCN2 and HCN4 were also reported to be the main subunits expressed in these neurons [5].

Finally, an important role of A-type potassium channels in controlling pacemaker frequency has been demonstrated [20,21]. After the AHP, I_A_ opposes the depolarization produced by inward currents to delay the triggering of the action potential. Single-cell PCR, in situ hybridization, and immunohistochemistry experiments suggested that I_A_ is carried exclusively by Kv4.3 channels [5,20,22,23,24]. Functional coupling between Kv4.3 and HCN channels was also revealed as the gating properties of the two channels were found to co-variate from cell to cell [21] via a common sensitivity to cytosolic cAMP and calcium levels. Furthermore, a co-varying gene module encompassing *Kcnd3*, *Kcnn3*, and *Scn2a* coding for Kv4.3, SK3, and Nav1.2, respectively, was revealed through single-cell microfluidic qPCR [24].

Our goal was to identify and quantify interactants of Cav1.2, Cav1.3, HCN2, HCN4, Kv4.3, and SK3 channels using the highly resolutive technique of LC-MS/MS mass spectrometry, which has contributed to the robust characterization of many supramolecular signaling complexes encompassing ion channels. Among other studies, Cav2 channels were found to be embedded into protein networks containing around ∼200 proteins [25], including ion channels, transporters, G-protein-coupled receptor-mediated signaling, and the release machinery of synaptic vesicles. More recently, the T-type Cav channel interactome [26] was depicted as a “T-type calcium channelosome”.

We first performed immunoprecipitation experiments followed by identification of co-purified channels using immunoblots. We uncovered a macromolecular complex comprising Cav1.3, HCN, and SK3 channels. Using LC-MS/MS mass spectrometry, we searched for network proteins that could organize this supramolecular signaling complex, based on their co-purification with Cav1.2, Cav1.3, HCN2, HCN4, Kv4.3, and SK3 channels. In addition to discovering several new ion-channel-interacting proteins, we confirmed the presence of the macromolecular complex composed of Cav1.3, HCN, and SK3 channel subunits. Exploiting in situ proximity ligation assay (PLA) imaging, we ascertained its presence in SNc DA neurons, suggesting that it could contribute to the robustness of the pacemaking activity of DA neurons.

## 2. Material and Methods

### 2.1. Animals

Wild-type (Wt) or KO mice aged 22 to 49 postnatal days were used. Breeding, husbandry, and the execution of experimental procedures were approved by the European Council Directive 86/609/EEC and the French National Research Council. All efforts were made to minimize the number of animals (reduce principle) and to provide the best breeding, accommodation conditions, and care necessary for the health and well-being of animals (refine principle). All mice and breeding colonies were maintained in animal facilities of the university under standard laboratory conditions, with an artificial 12 h light/dark cycle with an ambient temperature and free access to food and water. Experiments employed male and female animals.

KO mice in the C57BL/6J background strain (The Jackson Laboratory, Bar Harbor, ME, USA) used were Cav1.3^−/−^ ([27], a gift from J. Striessnig), Kv4.3^−/−^ (Deltagen, San Mateo, CA, USA), and Sarm1^−/−^ (The Jackson Laboratory). The genotype was determined through PCR experiments.

Mice were anesthetized using an induction chamber (Tem Sega, Pessac, France) with 100% oxygen flowing (3 L/min) through an isoflurane (Iso-Vet, Paris, France) vaporizer assuring deep anesthesia. Brains were dissected immediately after decapitation.

### 2.2. Membrane Solubilization

First, 5 or 10 (in Cav1.3 immunoprecipitation experiments) midbrains were quickly dissected on ice, collected at 4 °C in low calcium ACSF (CaCl_2_ 0.5 mM; MgCl_2_ 4 mM) with 10 mM of Hepes and 25 mM of glucose, and pooled. Using a smooth Teflon pestle in a tissue grinder of 0.08–0.13 mm clearance (3431E10, Thomas Scientific, Swedesboro, NJ, USA), brains were then manually homogenized in 10% (*w*/*v*) sucrose 0.32 M, Hepes 10 mM, 1 mM EDTA, pH 7.4 buffer with mammalian protease inhibitor cocktail (Sigma-Aldrich, Saint-Louis, MO, USA. #P8340) at 4 °C. Plasma-membrane-enriched protein fractions were prepared [28]. Nuclei and cell debris were discarded in the pellet after centrifugation at 900× *g* for 10 min. Membranes were then collected in the pellet after centrifugation of the supernatant at 10,000× *g* for 20 min. The pellet was resuspended with sucrose buffer, and colorimetric Bradford protein assay (Coomassie Plus, Thermo Fischer Scientific, Waltham, MA, USA) was performed to evaluate protein concentration.

### 2.3. Antibodies

All antibodies are listed in Table 1 with their corresponding RRID and concentrations used in the different protocols.

### 2.4. Protein Co-Immunoprecipitation Followed by Immunoblot

Plasma-membrane-enriched protein fractions from midbrain (0.7 to 1.7 mg protein) were solubilized for 30 min at 4 °C using the ComplexioLyte (Logopharm, GmbH, March, Germany) CL47a, CL48, CL60, CL80, CL87a, CL91, CL99, and CL120 buffers of varying stringencies at a concentration of 1.5 mg protein/mL of solubilization buffer with protease inhibitors and 1 mM of EDTA. Soluble material was recovered in the supernatant after ultracentrifugation for 10 min at 150,000× *g*. About 5 μg of antibodies per mg of proteins was added to the extract and incubated overnight at 4 °C. Antibody complexes were captured on protein A–Sepharose CL-4B or protein A/G (Santa Cruz Biotechnology, Dallas, TX, USA) and washed twice in TBS with 0.1% (*v*/*v*) ComplexioLyte.

### 2.5. Immunoblots

First, 40 µg of proteins or protein A–Sepharose bead pellets were resolved using SDS page on precast polyacrylamide gel (Biorad, Hercules, CA, USA), blotted on PVDF membrane (iBlot gel transfer stacks, Thermo Fisher Scientific, Waltham, MA, USA). Membranes were blocked in 0.2% (*w*/*v*) I-Block blocking solution (Thermo Fisher Scientific) prepared in Tris-saline buffer (TBS) and incubated overnight at 4 °C with primary antibodies (see Table 1) diluted in I-Block buffer. PVDF membranes were washed in TBS, 0.05% Tween 20 (*v*/*v*). Detection of primary antibodies was performed by incubating blots with 0.8 ng/mL of peroxidase-conjugated goat anti-rabbit (#711-035-152; Jackson ImmunoResearch) or anti-mouse (#115-035-068; Jackson ImmunoResearch) antibodies. IgGs were washed with the same buffer and subjected to enhanced chemiluminescence detection (ECL prime, GE Healthcare, Chicago, IL, USA). The blots were then imaged with the Azure Biosystems’s imager (Dublin, CA, USA).

### 2.6. Immunoaffinity Purification Followed by LC-MS/MS

The CL47a ComplexioLyte (Logopharm, GmbH, March, Germany) solubilization buffer, able to efficiently extract the six channel subunits from the lipidic membrane minimizing protein complex disruption, was chosen. Solubilization was carried out at a concentration of 1.25 mg protein/mL of solubilization buffer with protease inhibitors and 1 mM of EDTA. Affinity purified antibodies against Cav1.2, Cav1.3, HCN2, HCN4, Kv4.3, and SK3 subunits were obtained from Alomone in BSA-free buffer. All antibodies except antibodies against α1Cav1.3 were batch-conjugated to M-270 Epoxy beads using the Dynabeads Antibody Coupling Kit (14311D, Thermo Fisher Scientific) with 200 µg of antibodies to coat 10 mg beads. Dynabeads coated with protein A (10002D, Thermo Fisher Scientific) were used for Cav1.3 imuunoprecipitation. Freshly prepared solubilisates (2 mL) were incubated for 2 h at 4 °C with 1.9 mg of antibody-coated magnetic beads. After 2 brief washes with 1000 × diluted solubilization solution in TBS buffer, bound proteins were eluted with Laemmli buffer (DTT added after elution). Eluates were denaturated with SDS and shortly run on a 5–15% gradient SDS/PAGE gel (Biorad), and protein bands were cut into 3 separate slices.

### 2.7. Mass Spectrometry

For LC–MS/MS analysis, each band was further cut into small 1 mm^3^ pieces and rinsed 3 times with H_2_O and then with ammonium bicarbonate (NH_4_HCO_3_ 100 mM) and acetonitrile. Gel slices were reduced with DTT (10 mM in 100 mM of NH_4_HCO_3_ for 1 h at 56 °C) and then alkylated with iodoacetamide (55 mM in 100 mM of NH_4_HCO_3_, for 30 min, in the dark). After discarding the supernatant, samples were shrunk with acetonitrile and supplemented with 5 µL of 12.5 ng/µL of porcine trypsin endopeptidase (Promega, Madison, WI, USA) and incubated overnight at 37 °C in 25 mM of NH_4_HCO_3_. The supernatant was transferred to a fresh tube. Subsequently, the gel was extracted 3 times with 50% acetonitrile (*v*/*v*) in H_2_O containing 0.1% (*v*/*v*) formic acid. The digests were evaporated to dryness in a speed vac. Before LC-MS/MS analysis, samples were reconstituted in 8 µL of 2% (*v*/*v*) acetonitrile with 0.1% (*v*/*v*) formic acid. An aliquot of each sample (6.4 µL, representing 80% of the total sample) was analyzed through LC–MS/MS. The nano high-performance liquid chromatography (HPLC) UltiMate 3000 RSLC nano system (Thermo Fisher Scientific, San Jose, CA, USA) coupled with a Q-Exactive mass spectrometer equipped with a Nanospray Flex ion source (Thermo Fisher Scientific, San Jose, CA, USA) was used to separate protein samples. Peptides were loaded onto a trap column (PepMap Acclaim C18, 5 mm × 300 μm ID, 5 μm particles, 100 Å pore size, Thermo Fisher Scientific, San Jose, CA, USA) at a flow rate of 20 µL/min using 0.1% (*v*/*v*) formic acid (solvant A) as a mobile phase. After 3 min, the trap column was switched in line with the analytical column (PepMap Acclaim C18, 15 cm × 75 μm ID, 2 μm, 100 Å, Thermo Fisher Scientific, San Jose, CA, USA). Peptides were eluted using a flow rate of 300 nL/min and a binary 90 min gradient. The gradient started with the mobile phases 98% A, 0.1% (*v*/*v*) formic acid in H_2_O, and 2% B, 0.1% (*v*/*v*) formic acid in acetonitrile, increased to 25% B over the next 60 min, followed by a steep gradient to 90% B for 6 min. After a 9 min hold, the gradient was ramped down over 1 min to the starting conditions of 98% A and 2% B for equilibration for 15 min at 30 °C. The Q-Exactive mass spectrometer was operated in data-dependent mode, using a full scan (*m*/*z* range 350–1800, nominal resolution of 35,000, target value 3 × 10^−6^), followed by tandem mass spectrometry (MS/MS) scans of the 10 most abundant ions (Top 10). MS/MS spectra were acquired in the centroid mode using normalized collision energy of 27, isolation width of 2.0 *m*/*z*, resolution of 17,500, target value of 2 × 10^−5^, and maximum fill time 120 ms. Precursor ions selected for fragmentation (charge states 2–6) were put on a dynamic exclusion list for 45 s. Additionally, the minimum AGC target was set to 2 × 10^−3^, and the intensity threshold was calculated to be 1.7 × 10^−4^. The peptide match feature was set to preferred, and the exclude isotopes feature was enabled. For peptide identification, the RAW-files were loaded into Proteome Discoverer (version 2.4, Thermo Fisher Scientific, San Jose, CA, USA) to generate Mascot generic files (.mgf). All created MS/MS spectra were searched using an in-house Mascot server version 2.3 (www.matrixscience.com, MatrixScience Ltd., London, UK. Mgf files were searched against the Mus Musculus Swiss-Prot database (17,070 sequences, version downloaded in June 2019) using the following search parameters: the peptide mass tolerance was set to ±10 ppm, and the fragment mass tolerance to ±0.2 Da. The maximal number of missed cleavages was set to 3, using tryptic specificity with no proline restriction. Carbamidomethyl on cysteine and oxidation on methionine was set as a variable modification. The minimum peptide length was set to 7 amino acids. These results were used for the quality control of IPs to confirm the successful immunoprecipitation of the bait compared to the corresponding control.

### 2.8. MS Data Analysis and Statistics

The resulting MS/MS data were processed using MaxQuant (v.1.6.17; downloaded from https://www.maxquant.org/, Cox lab, Max Planck institute of Biochemistry, Martinsried, Germany) to quantify peptide areas [29]. Proteins were quantified by summing unique and razor peptides. Tandem mass spectra were searched against the Uniprot Mus Musculus (20,205 entries, download at 17 June 2017) database concatenated with the reverse decoy database. Trypsin/P was specified as a cleavage enzyme, allowing up to two missing cleavages. Mass error was set to 10 ppm for precursor ions and 0.02 Da for fragment ions. Carbamidomethylation on cysteine was specified as fixed modification, and oxidation on Met and acetylation on protein N-terminal were specified as variable modifications. False discovery rate (FDR) thresholds for the protein, peptide, and modification site were specified at 1%. The minimum peptide length was set to 7, and “match between runs” was disabled to avoid misidentification [30]. Normalization was set to none. At least 2 unique peptides per protein group was required for the identification of the proteins. All of the other parameters in MaxQuant were set to default values. One MaxQuant analysis was computed for all samples.

MaxQuant search results were exported to the Perseus platform (version 1.6.15, https://maxquant.net/perseus/) for statistical analysis. The data transformation was adapted from the general proteomics workflow described in [31]. Protein quantification was based on the intensity-based absolute quantification (iBAQ) algorithm integrated into the MaxQuant platform. Reverse and “only identified by site” identifications were excluded from further data analysis. After log2 transformation of the leftover proteins, iBAQ values were normalized using Z-score, in which the median of each sample column was subtracted from each value and then divided by the standard deviation of the column. Normal distribution was then checked graphically. Contaminants were then filtered out of the identified protein list, and proteins were kept if at least 66% valid values were present in at least one protein group (control or baits). Missing values were then imputed assuming a normal distribution of 0.3 width and 3 downshifts. Significant data points were determined using a 200-permutation-based FDR with a Pearson correlation function using the Hawaii plot tool (multiple volcano plots) implemented in Perseus. Statistical testing using the default parameters allowed us to determine positive interactants classified into Class A (1% FDR, S0 = 0.1) or Class B (5% FDR, S0 = 0.1) [32]. For validation analysis, interactants were then compared with a contaminant repository for affinity purification (the CRAPome 2.0 database; [33]), collecting negative controls from multiple affinity purification MS studies. Only proteins identified in less than 1% of all CRAPome-negative experiments were considered highly specific interactants. The CRAPome values of interacting proteins are available in Appendix A. All Class-A-specific interactant proteins (cited as less than 1% of all CRAPome-negative experiments) were taken into account, although some Class A interactants detected in more than 1% of all CRAPome negative experiments or Class B proteins were considered because of their potential interest or known interactions in the literature. In summary, 43 IPs were analyzed in nano-LC MS/MS runs that identify an average of 3400 proteins among all affinity purifications, with 900 proteins quantified.

The mass spectrometry proteomics data have been deposited to the ProteomeXchange consortium via the Pride Proteomics Identification Database partner repository [34], with the data set identifier PXD#034883. The processed proteomics data are available in the Appendix A.

### 2.9. Proximity Ligation In Situ Assay (PLA)

Mice were deeply anesthetized in a box with a flow of 5% isoflurane and then transcardially perfused with ice-cold PBS and paraformaldehyde 4% (*v*/*v*) in PBS under a constant flow of 2.5% isoflurane.

Brains were removed and incubated for 2 h in the same fixative solution and kept overnight in PBS at 4 °C. Then, 40 µm coronal midbrain sections were sliced using a vibratome (vibrating microtome 7000 smz, Campden Instruments, Loughborough, UK) and collected as floating sections at 4 °C.

For each independent experiment, negative controls were performed on the same day using either slices from KO mice deprived of one of the two antigens recognized by the dual primary antibodies or using an antibody recognizing a protein not expressed or not in tight proximity of the protein of interest.

Most of the slices were prepared for antigen retrieval by heating at 80 °C for 30 min in 10 mM of sodium citrate (Sigma-Aldrich, Saint-Louis, MO, USA). Dual-antigen recognition PLA (Duolink, Sigma-Aldrich, Saint-Louis, MO, USA) experiments were conducted according to the manufacturer’s instructions with the following modifications: incubation with blocking solution, PLA probes, and ligation steps were extended to 1 h, 2 h and 45 min, respectively; the amplification step consisted of a 2 h incubation at 37 °C with a 1/60 concentration of polymerase. Dual primary antibodies used are described in Table 1. Subsequent secondary labeling was accomplished with the use of anti-rabbit and anti-mouse PLA probes (Sigma-Aldrich). To detect the PLA signal, green reagent (Sigma-Aldrich, DUO92014) was used. Dopaminergic neurons were successively immunolabeled using chicken anti-tyrosine hydroxylase antibodies (Abcam # ab76442) followed by Alexa-Fluor^®^-594-conjugated goat anti-chicken IgGs (#103-585-155; Jackson ImmunoResearch, West Grove, PA, USA). Slices were then mounted in Dako Faramount aqueous medium (Agilent, Santa Clara, CA, USA).

### 2.10. Image Analysis/Quantification of PLA Signal

High-resolution (63X 1.4 NA) images from single scans were analyzed in ImageJ (NIH) to count the number of PLA puncta. A threshold was selected manually to discriminate PLA puncta from background fluorescence. Once selected, this threshold was applied uniformly to all images in the sample set. The ImageJ (version 1.53h) “analyze particle” function was used to count particles larger than 0.1 μm^2^ for the PLA signal. A mask obtained from tyrosine-hydroxylase-labeled DA soma was used to select PLA particles specifically located on DA neurons. For each experiment and condition, several randomly selected non-overlapping vision fields were analyzed. Data from different sample sets were pooled, and Wilcoxon–Mann–Whitney tests were conducted using SigmaPlot 10.00 (Grafiti LLC, Palo-Alto, CA, USA) to compare PLA signals between samples and negative controls. Also, stratified Mann–Whitney tests were performed to analyze the distinct sets of experiments and to provide even more robust significance.

## 3. Results

Optimization of extraction and purification conditions were individually searched for the six channel protein complexes testing eight solubilization buffers of varying stringencies from Logopharm (GmbH, March, Germany). The solubilization buffer CL47a, able to efficiently extract the six channel subunits from the lipidic membrane, was selected to perform co-immunoprecipitation experiments.

### 3.1. Cav1.3-SK3-HCN Channel Complex Detection

As a first step in our investigation of the molecular interactions between somatodendritic ion channels expressed in SNc DA neurons, we decided to use ion channel immunoprecipitation with specific antibodies and subsequent detection on immunoblots. Based on the literature, we focused on the potential interactions between Cav1.3, Kv4.3, HCN2, HCN4, and SK3 channels. Mouse midbrain plasma-membrane-enriched protein fractions were used as the starting material (Figure 1A,B, lane P2). HCN4 channel subunits were co-immunoprecipitated with Cav1.3 (Figure 1A, left panel, lane IP1) and SK3 (Figure 1B, left panel, lane IP1) channel proteins in buffer CL47a. HCN4 co-purifications were found in all solubilization buffers of varying stringencies tested, suggesting strong interactions of HCN4 with Cav1.3 and HCN4 with SK3 channels. Furthermore, SK3 proteins were immunoprecipitated with anti-Cav1.3 antibodies (Figure 1A, right panel) and Cav1.3 subunits with anti-SK3 antibodies (Figure 1B, right panel) in the CL47a solubilization buffer. No molecular interactions were observed between Kv4.3 and HCN2 or HCN4 channels (Figure 1C, middle and right panels, respectively). Although HCN2 (Figure 1C, middle panel) and HCN4 (Figure 1C, right panel) channels were observed in the solubilized starting material (sol) before immunoprecipitation, they were not co-purified with Kv4.3 channels immunoprecipitated in buffers CL47a (Figure 1C, lane IP1) or CL48 (Figure 1C, lane IP2), the most powerful buffer for Kv4.3 channel extraction. The same results were obtained in all solubilization buffers of varying stringencies tested. However, we cannot exclude any binding of Kv4.3 channels with the identified macromolecular complex as Kv4.3 proteins appear more difficult to extract from the membranes than the other channels (see the absence of Kv4.3 labeling in the solubilized starting material in Figure 1C, left panel, lane sol). Furthermore, Kv4.3-immunolabeled bands of larger apparent molecular weights were observed, possibly indicating the presence of Kv4.3 channels in large partially denaturated complexes.

The identified channel co-immunoprecipitations revealed through immunoblot are summarized in Figure 2.

### 3.2. Identification of Proteins Co-Purified with Midbrain Ion Channels Using LC-MS/MS Mass Spectrometry

After the targeted immunoprecipitation experiments, we carried out an unbiased screening approach to identify the proteins interacting with the six ion channels of interest, using co-immunoprecipitation combined with mass spectrometry (the experimental workflow is summarized in Figure 3).

In order to minimize false-positive identification of background proteins while not extra-filtering proteins specifically recognized by antibodies, we used magnetic beads as a support for immunoprecipitation as their reduced size, hydrophobicity, and porosity have an effective, positive impact on purification specificity [35]. Also, short antibody incubations of only 2 h were used as many studies have demonstrated that rapid purifications ensure specific interactant enrichment [25,36,37].

We also incorporated many proper negative controls. We thus performed experiments on target knock-out tissues whenever it was possible. As most of our targets are ion channels essential for brain and heart functions, it was not possible to produce KO mice for each chosen target. We performed four independent IPs using anti-Kv4.3 antibodies on Kv4.3^−/−^ KO mice. We also raised Cav1.3^−/−^ KO mice, but due to the poor health of these animals and low detection of Cav1.3 channels, we could not perform MS/MS experiments. So, we included seven IPs using antibody against an unrelated target, Sarm1, on Sarm1^−/−^ KO membranes. Although Sarm1 was not one of the targeted ion channels, most non-specific interacting proteins were assigned in these negative controls.

Our large study using, in parallel, seven antibodies targeting different baits led us to discriminate unspecific binding, as different antibodies are less likely to cross-react with the same background proteins. Furthermore, many biological replicates of immunoprecipitations (IPs), key to the identification of robust interactions, were performed on wild-type mice based on four, four, three, six, seven, and eight replicates of Cav1.2, Cav1.3, HCN2, HCN4, Kv4.3, and SK3 subunits, respectively, for a total of forty-three independent immunopurifications.

Multiple research groups have noticed that the best way to characterize specific interactions using MS/MS is to monitor quantitative enrichment in the mass spectrometer. Therefore, we measured the intensity of peptide ions corresponding to a given interactant (the prey) in the purification of the bait and compared it to the “noise” corresponding to the same prey in the control analyses. This gives the possibility to measure enrichment in the mass spectrometer and to discriminate false-positive data, as true interactants are enriched throughout purification. Statistical comparisons of all of the data from IP experiments with a probabilistic method using Perseus software (version 1.6.15, https://maxquant.net/perseus/, Cox lab, Max Planck institute of Biochemistry, Martinsried, Germany) gave high confidence to detect significant interactants.

Finally, the use of contaminant repository data for affinity purification (CRAPome) made of negative controls generated by the proteomics research community allowed for the capture of the most complete set of contaminants. That strategy is based on the shared observations through large data samples that negative controls are largely bait-independent [33].

As an internal validation, we identified several ion channel isoforms as well as auxiliary subunits and already-published interactants of the ion channels included in our study, thus further confirming the efficiency of our strategy.

#### 3.2.1. Characterization of the L-Type Calcium Channel Complexes

Proteins co-purifying with Cav1.2 channel subunits were identified (Figure 4A, left panel). The well-known voltage-dependent calcium channel auxiliary subunits β1-4 (*Cacnb1-4* genes) and α2δ1-2 (*Cacn2d1-2* genes) were detected at a significant level. In addition, the Ca^2+^-calmodulin-activated phosphatase calcineurin A (*Ppp3ca* gene), the G-protein-activated inward rectifier potassium channels 1 and 2 (girk1 and girk2, also named Kir3.1 (*Kcnj3* gene) and Kir3.2 (*Kcnj6* gene)), and the adenylate cyclase 1 (*Adcy1* gene) were identified. Unexpectedly, we also found a highly significant interaction of α_1_-Cav1.2 subunit with Pex5, a protein that associates type 1 peroxisomal targeting signal (PTS1)-containing proteins [38] with peroxisome membranes.

Proteins co-purifying with Cav1.3 channel subunits are shown on the right panel of Figure 4A. They were mostly distinct from the ones found for Cav1.2 channels. It is especially noticeable that no voltage-dependent calcium channel auxiliary subunits were significantly co-purified with the pore-forming α1 protein (Figure 4A and Figure 5). Furthermore, interactions with HCN2, HCN4 channel subunits, and Pex5-related protein Pex5r (*Pex5l* gene) were found, confirming our results of co-immunoprecipitation and immunoblot detection. Also, Rab-interacting molecule-binding protein Rimbp2 promoting the clustering of Cav1.3 channels at the active zone [39] was found, albeit with a low significance level (Figure 5). The Rimbp2 binding interactant Munc 13 (*Unc13c* gene) [40], the mouse orthologue of unc-13, was also identified (Figure 5).

#### 3.2.2. Characterization of the SK3 Potassium Channel Complex

Many highly significant interacting proteins were found to co-purify with SK3 (*Kcnn3* gene) proteins (Figure 4B). SK2 (*Kcnn2* gene) channel subunits were detected, pointing out the possibility of SK channel heterodimer formation in midbrain neurons. Surprisingly, SK3 subunit interactions were detected with the sclerosis tuberous complex (Tsc) proteins Tsc1, Tsc2, and Tbc1d7, inhibitors of the mechanistic target of rapamycin (mTOR) serine/threonine kinase. Tsc1 was the highest significant binding protein of the SK3 complex. Remarkably, SK3 subunits were found to interact with several calcium-related proteins, such as the α1 subunit of R-type voltage-dependent calcium channel or Cav2.3 (*Cacna1e* gene) and α2δ1 (*Cacna2d1* gene) subunits, as well as the membrane-specific calcium/calmodulin-dependent serine kinase Cask protein, a member of the CAMKII family. The A-kinase anchoring protein 5 (Akap5) was also found. Interestingly, HCN2 channels were found to co-immunoprecipitate with SK3 channels, confirming the interactions detected on immunoblots.

#### 3.2.3. Characterization of the HCN Channel Complexes

HCN2 channels were immunoprecipitated with HCN auxiliary subunit Pex5r (*Pex5l* gene), also named Rab8b-interacting protein (Trip8b) (Figure 4C, left panel). Moreover, HCN1 and HCN4 were co-purified with HCN2 channels. The voltage-gated potassium channel Kv3.3 (*Kcnc3* gene) was also identified. Furthermore, glutamate ionotropic receptor AMPA (AMPAR) type subunit 2 GluR2 (*Gria2* gene) together with Frrs1l and Cpt1c known to assemble in AMPAR subcomplex were co-purified with HCN2 channel subunits.

Although HCN4 seems to form heteromers with HCN2 and HCN1 subunits (Figure 4C, right panel), the characterized interacting protein partners of HCN4 were different from those found for HCN2. For instance, we found that sterile alpha and Toll/interleukin-1 receptor (TIR) motif-containing 1 (Sarm1) was a robust and significant binding partner of HCN4 subunits (Figure 4C, right panel), while it was absent from HCN2 immunoprecipitation (Figure 4C, left panel).

To understand the difference in binding partners between HCN2 and HCN4 channels, we studied HCN2, HCN4 channel, and Sarm1 protein expression during development. The two HCN subunits displayed opposite developmental trajectories, with an age-related increase in HCN2 protein expression (Figure 6, left panel) and a reciprocal age-related reduction of HCN4 (Figure 6, middle panel). Consistent with its demonstrated molecular interactions with HCN4 (Figure 4C), Sarm1 protein displayed a sharp developmental expression profile, being expressed in juvenile mice (P9) but barely detectable in adult midbrain neuronal membranes (Figure 6, right panel).

#### 3.2.4. Characterization of the Kv4.3 Potassium Channel Complex

As shown on the Hawaii plot (Figure 4D), the known Kv4.3 channel auxiliary subunits Kchip 4 (*Kcnip4*), Dpp6 and Dpp10 were identified as well as Kv4.2 (*Kcnd2*) channels, which can form heterodimers with Kv4.3 subunits. No other significant molecular interactants were found, maybe partly due to the low extraction of Kv4.3 channels from midbrain membranes.

### 3.3. Identification of the Cav1.3-SK3-HCN Complex in SNc DA Neurons Using Proximity Ligation Assay

The proximity ligation assay (PLA) method allows interacting proteins to be spatially and quantitatively visualized using two different primary antibodies against each of the two proteins of interest followed by a single amplification process only taking place if the two proteins are in close proximity to each other. Using in situ PLA imaging in wild-type and Cav1.3^−/−^ mice (negative control), we could identify specific interactions between Cav1.3 and HCN2 in SNc DA neurons, using tyrosine hydroxylase immunolabeling to isolate interactions specifically in DA neurons (Figure 7A,C left panel).

Also, SK3 subunits were found in close proximity to HCN2 proteins (Figure 7B, top row and C middle panel), but not SK2 (Figure 7B, bottom row and C middle panel) or Kv4.3 (Figure 7C, middle panel) subunits. PLA analysis revealed that Cav1.3 and SK3 proteins were also in close proximity to SNc DA neurons (Figure 7C, right panel), strengthening the observation that HCN2 interactions depicted using anti-Cav1.3 and anti-SK3 channel antibodies were not due to antibody cross-reactivity.

Additionally, we could validate an interaction between SK3 and Tsc1 (Figure 8) and between HCN4 and Sarm1 (Figure 9) in SNc DA neurons using PLA.

The molecular interactions identified using the PLA technique in SNc DA neurons compared with those established through co-immunoprecipitation followed by immunoblot or by mass spectrometry analyses on midbrain membranes are schematically shown in Figure 10.

## 4. Discussion

Using high-resolution proteomics on midbrain tissues (mass spectrometry in particular) and immunohistochemistry (proximity ligation assay) techniques, we produced a large screening of interacting proteins linking several somatodendritic ion channels involved in the generation and regulation of pacemaking activity in SNc DA neurons. Several new interactants were discovered, and a macromolecular complex encompassing Cav1.3, SK3, and HCN channels was characterized.

### 4.1. Limitations and Perspectives

While we have taken great care to minimize any potential unspecific interactions, determining true interactants solely using immunoprecipitation coupled to mass spectrometry is a challenging task. Therefore, further verifications should be conducted to definitively exclude any potential antibody-related artifacts, such as cross-reactivity of certain antibodies. One approach could be to utilize multiple different antibodies targeting the same ion channel and to consider only the common interactants identified across these various immunoprecipitations. But, undoubtedly, the best option would be to introduce additional negative controls using conditional knock-out mice with substantia nigra dopaminergic neuron deleted of HCN2, HCN4, SK3, Cav1.2, and Cav1.3 at an adult stage to obtain sufficient midbrain membrane material. Such stringent control would serve to corroborate the interactions outlined in this article, thereby enhancing the reliability and validity of our findings.

### 4.2. L-Type Calcium Channel Interactants

Cav1.2 channel α1-subunits were found to interact with voltage-dependent calcium channel auxiliary subunits β1-4 and α2δ1-2. Also, proteins of the β2 adrenergic receptor (β2AR) supramolecular complex, reported to be involved in the sympathetic control of heart performance (see [41] for a review), were identified: Ca^2+^-calmodulin-activated phosphatase calcineurin A and adenylate cyclase. The G protein-activated inward rectifier potassium channels Girk2 were also observed. They are expressed in most SNc DA neurons [42], where they interact in vitro with the dopamine D2 receptor (D2R) and adenylate cyclase [43] to mediate D2R auto-inhibition of cell firing [44]. So, these physical interactions within a Cav1.2 channel complex could support rapid short-term modulation of Ca^2+^ entry under the control of G-protein-coupled receptors. Interestingly, a calcineurin-dependent pathway is involved in α-synuclein-induced degeneration of midbrain dopaminergic neurons [45].

Unexpectedly, we also found a highly significant interaction between α1-Cav1.2 subunits and Pex5, a protein that associates the peroxisome membrane with type 1 peroxisomal targeting signal (PTS1)-containing proteins [46]. Pex5 shows a high homology with Pex5r or Trip8b, the HCN channel accessory subunits, the binding partner of the vesicle trafficking regulator Rab8b [47] and of latrophilin (the α-latrotoxin receptor) [48]. Pex5 and Pex5r both associate with the consensus carboxyl-terminal PTS1 motif [46], which is, for the mouse HCN1, HCN2, and HCN4 subunits, serine-asparagine-leucine. Remarkably, the carboxyl-terminus of Cav1.2 subunit contains an identical amino acid motif.

In contrast with Cav1.2, and although Cav1.3 channels have been described to be the major voltage-dependent calcium channels in SNc neurons [4], a poor rate of Cav1.3 subunit purification was obtained from midbrain plasma membranes. As Cav1.3 channel distribution is strictly regulated at the plasma membrane in pancreatic β-cells [49], cardiomyocyte [50], inner hair cells [51], and HEK-293 cells expressing Cav1.3 channels [52], it is likely that Cav1.3 subunits were present at low concentrations in the midbrain plasma membrane-enriched fraction. In that sense, the voltage-dependent calcium channel auxiliary subunits known to be essential to membrane localization Cavβ [53,54] and α2δ1 [55] were not significantly detected in the Cav1.3 complex. Noticeably, the Rab-interacting molecule-binding protein Rimbp2 was found in the Cav1.3 nano-environment. These two proteins were described to be in close association at the presynaptic active zone of ribbon synapses of inner hair cells [56], where Rimbp2 was shown to stabilize Cav1.3 channel inactivation properties essential for sound-induced tonic neurotransmitter release [39].

### 4.3. SK Channel Interactants

Major interactants of SK3 subunits identified through MS-MS mass spectrometry were the tuberous sclerosis complex Tsc1 and Tsc2 proteins. Furthermore, we observed SK3–Tsc1 potential interactions in SNc neurons using in situ PLA imaging. Tsc proteins are inhibitors of mTOR, which has been linked to the regulation of voltage-dependent Kv1.1 [57,58,59] and Kv4.2 [60] potassium channels. Moreover, in pyramidal neurons expressing a constitutively active mTOR, an ectopic synthesis of HCN4 channel was observed [61]. In Tsc2^−/−^ human differentiated excitatory cortical neurons [62], neuronal activity was increased, with highly synchronized Ca^2+^ spikes and an enhanced expression of Cav1.3. It is interesting to note that Kosillo et al. [63] showed a significant decrease in the firing frequency of SNc DA neurons of Tsc1^−/−^ animals, which could potentially be due to a higher SK channel activity.

In addition, SK3 channels were found to be intricately associated with calcium-related proteins, such as the voltage-dependent calcium channel α1-R-type and α2δ1 subunits. In SNc DA neurons, R-type Ca^2+^ currents participate in somatodendritic Ca^2+^ oscillations [64]. This activity may contribute to the selective degeneration of these neurons in Parkinson’s Disease, as deletion of the Cav2.3 gene is neuroprotective in a mouse model of the disease. Furthermore, in SNc DA neurons of Cav2.3 KO mice, Ca^2+^-dependent AHPs carried by SK channels were also significantly reduced [64]. Akap5, known to regulate L-type calcium channel phosphorylation through local anchoring of PKA [65], was also found to be associated with SK3 channels. Also, membrane-specific Cask protein was identified in the SK3 protein complex. This PDZ scaffolding protein coordinates a large number of targets through multiple interactions together with Mint1 and amyloid-beta A4 precursor protein-binding family A member 1 (*Apba1*), which was also detected as an SK3 interactant. Interactions of Cask with Cav2.2 [66], Kir2 [67], and Nav1.5 [68] channels, as well as NMDA [69] and ACh receptors [70], were reported. Furthermore, Cask was selected as a new candidate gene for understanding Parkinson’s Disease via its co-expression with E3 ubiquitin-protein ligase parkin in microarray data of induced pluripotent stem cells derived from patients [71].

### 4.4. HCN Channel Interactants

When HCN2 channels were immunoprecipitated with specific antibodies, HCN1, HCN3, HCN4, and Pex5r subunits were found, pointing out the probable heteromerization of HCN subunits in midbrain neurons. The presence of HCN1 was not expected, as co-expression of HCN2, HCN3, and HCN4 but not HCN1 mRNA was observed in single-neuron multiplexed PCR performed on SNc neurons, which correlated with the absence of fast gating for the I_H_ current [19]. Therefore, it is highly likely that HCN1 detection in our study arises from non-dopaminergic midbrain neurons, including, for instance, substantia nigra GABAergic neurons. Consistent with this hypothesis, substantia nigra HCN1 transcripts are reported in the genotype–tissue expression project database report (dbGaP Accession phs000424.v8.p2), and equivalent densities of HCN1, HCN2, and HCN4 proteins were revealed through immunostaining in neurons of the substantia nigra [72]. The voltage-gated potassium channel Kv3.3 (*Kcnc3* gene) was identified as a highly significant interactant. Interestingly, Kv3.3 mRNA and proteins are expressed in midbrain neurons [73,74]. They belong to the family of Kv3 channels, which open only during action potentials, contributing to fast repolarization [75] and energy-efficient firing [76]. GluR2 (*Gria2* gene) constituting AMPAR was co-purified with HCN2 channel subunits. This physical co-presence in a molecular complex would provide a morphological basis for excitatory glutamatergic synaptic input integration observed in a variety of neurons (see [77] for a review). Indeed, HCN2 was found to co-localize with GluR4 (*Gria4*) in dendritic spines of reticular thalamic neurons [78]. Also, co-localization of HCN and GluR channels was described by immunogold-electron microscopic labeling in spine-like microvilli sensory neurons [79]. Furthermore, functional interactions were also suggested by the fact that I_H_ currents were specifically increased in the epileptic stargazer mouse [80] bearing a spontaneous mutation of the AMPAR auxiliary subunit (*Cacng2*) gene [81].

Interacting protein partners of HCN4 were markedly different from those found for HCN2, even though HCN4 seems to form heteromers with HCN2 and HCN1 subunits. This result is in line with HCN2 and HCN4 channel differences in function, distribution, and regulation. HCN4 channels are mostly responsible for the I_f_ current of the sinoatrial node, the pacemaker region of the heart, generating intrinsic heartbeat (see [82] for a review) and regulating oscillation activity in thalamocortical neurons. Substantial differences in cellular localization have been observed for HCN2 and HCN4. While HCN2 are mostly detected in dendrites in mature rat hippocampal (see [83] for a review) and reticular thalamic neurons [84], HCN4 channels are mainly observed in the soma, although no differences were noticed in SNc DA neurons [5]. Also, HCN4 channels seem to play a specific role early in development. They display an early expression in human and mouse embryonic cells, where they are involved in right–left asymmetry patterning, only at an early stage [85]. They affect the development of the olfactory projection map in mice through spontaneous firing rate regulation [86]. Interestingly, we could reveal in midbrain neurons a differential developmental profile for HCN2 and HCN4 channel proteins consistent with what was found in the development of hippocampal [87] and cortical [88] neurons, i.e., a predominant expression of HCN4 in early developmental stages.

We found that sterile alpha and Toll/interleukin-1 receptor (TIR) motif-containing 1 (Sarm1) was a robust and significant binding partner of HCN4 subunits. *Sarm1* knockdown confers neuroprotection to axon degeneration in Drosophila and mice [89] due to a nicotinamide adenine dinucleotide (oxidized form, NAD^+^) cleaving enzymatic activity (for a review, see [90]). We established that Sarm1 displays a developmental expression similar to HCN4 in midbrain neurons, being prominent early in development, as seen in the whole brain [91]. Sarm1 plays a role in the regulation of signaling pathways that influence dendritic arborization and local axon branch pruning [92]. Remarkably, Tir-1, the Sarm1-ortholog in *C. elegans*, shares with HCN4 an early developmental role in left–right asymmetric signaling [93], strictly dependent on voltage-activated N/PQ- and the L-type Ca^2+^ channel, Ca^2+^-activated K^+^, and cyclic nucleotide-gated channel activity [94].

### 4.5. The Surprising Absence of Interactants for Kv4.3

Kv4.3 channels, however, appeared isolated, as the only identified interactions were with Kv4.2 subunits and with the auxiliary subunits Kchip and Dpp proteins. As several reports indicate a subtle interplay between the Kv4.3-mediated I_A_ current [20] and several inward currents in SNc DA neuron activity (see [1] for a review), the lack of interactions between Kv4 channels and other somatodendritic ion channels is especially surprising. Furthermore, in a previous report exploring the biophysical properties of SNc neuron currents, we observed a co-variation of the voltage dependences of I_A_ and I_H_ currents sensitive to calcium and cAMP levels [21]. Another study from our group also described a co-expression module comprising the genes encoding Kv4.3, SK3, and Nav1.2 channels [24]. The difficulty of extracting Kv4.3 proteins as single monomers from midbrain plasma membranes may partly explain the discrepancies between these functional studies and our biochemical investigation, even though our approach identified expected and relevant Kv4.3 interactants (in particular, auxiliary subunits). Larger-molecular-weight complexes were observed on immunoblots. As lipid 2-arachidonoylglycerol endocannabinoids were reported to regulate A-type channels in midbrain dopamine neurons [95] and Kv4.3 channels in cardiac myocytes [96], it is possible that a lipidic hydrophobic environment in the plasma membrane made protein solubilization and subsequent precipitation particularly difficult. Nevertheless, using in situ PLA imaging, we did not observe any proximity between SK3 and Kv4.3 channels in SNc neurons. Thus, although it is difficult to draw definitive conclusions about this ion channel, it appears that it has fairly scarce molecular interactions with the other somatodendritic ion channels that we investigated, in contrast with its tight functional interactions during pacemaking.

### 4.6. Putative Physiological and Physiopathological Meaning of the Ion Channel Partnerships

As summarized in Figure 10, interactions determined here depict that several ion channels were found to display molecular links in SNc DA neurons: Cav1.3, HCN2, HCN4, and SK3. Surprisingly, no molecular interactions with Kv4.3 were observed. As already extensively commented, all of these ion channels have several features in common: (i) they are all mainly located in the somatodendritic compartment where (ii) their role is to control the pace and regularity of the spontaneous activity of SNc DA neurons. The question that arises, then, is how this molecular organization might affect the functional interactions between these ion channels. As already mentioned, in spite of the co-variation of voltage dependences of IA and IH currents [21], Kv4.3 and HCN channels were not found to interact at the molecular level, meaning that co-regulation by the same signaling pathways may not imply a strict nanometric co-localization of channels. Interestingly, concerning HCN and Cav1.3 ion channels, one study suggested that pacemaking might rely on the synergistic actions of HCN, Cav1.3, and Nav channels, such that, in the absence of Cav1.3, pacemaking might rejuvenate to an HCN/Nav-dependent mechanism [7]. Although we did not investigate the potential interactions between Nav, HCN, and Cav1.3 channels in the current study, it is tempting to speculate that the co-localization of these channels may partly underlie their synergistic effects on pacemaking. Concerning Cav1.3 and SK3, several studies investigated the calcium sources responsible for SK channel activation in SNc DA neurons [16,97,98]. In two of these studies [16,98], the influence of L-type calcium (Cav1) channels to activate apamin-sensitive (SK) currents was tested, and both studies concluded that Cav1 channels do not support the calcium entry necessary for SK channel activation, although several types of calcium channels (including different subtypes of Cav2 and Cav3 channels) appear to mediate it [16,97,98]. Thus, the presence of biochemical interactions of Cav1.3 with SK3 contrasts with the demonstrated lack of dependence of SK3 activation on Cav1-mediated calcium entry. In one of these studies, though [98], Cav1 channels (but not Cav2 or Cav3 channels) were demonstrated to have an influence on firing regularity in SNc DA neurons, similar to the one observed for SK channels [5,16,98]. Thus, the functional data obtained so far on IH, IA, Ca2+ currents, and SK currents (AHP) do not appear consistent with the identified co-localization of channels, as co-regulated channels (HCN and Kv4.3) are not part of the same complex [21], and channels belonging to the same complex (Cav1.3 and SK3) do not seem to functionally interact [16,98]. In a previous study, we demonstrated that several ion channels participating in the control of pacemaking are co-regulated at the genetic level [24]. We postulated that the existence of such a co-expression module ensures that all of the ion channels critically involved in pacemaking are expressed altogether, participating in the robustness of the pacemaking process. Similarly, one may argue that the existence of molecular links between Cav1.3, HCN, and SK3 channels ensures that all of these ion channels will be addressed in the same appropriate compartment (most likely the somatodendritic compartment) and thus participate in the robustness of spontaneous activity.

Understanding the role of the ion channel complex in the pathophysiology of neurological disease is still in its infancy and lagging behind the comprehension of cardiac dysfunctions. Recently, Maurya et al. [99] identified protein interactants for 13 types of ion channels and established ion channel networks in murine and human heart tissues. Correlations were found in human population genetics data to specifically identify interacting proteins that influence the electrocardiogram [99]. Nevertheless, in neurons, AMPAR interacting partners regulating the number and function of AMPA receptors at the post-synapse, controlling synaptic strength and plasticity, have been often found in human patients suffering from schizophrenia or autistic spectrum disorders (see [100] for a review). Additionally, the AMPAR complex-interacting Frrs1l protein was shown to cause severe intellectual disability with cognitive impairment, speech delay, and epileptic activity [101]. Furthermore, Niemann–Pick Type C disease was depicted in an animal model as a nanostructural ion channel clustering disease, characterized by alterations in Kv2.1–Cav1.2 nanodomains, which contribute to neurodegeneration [102].

## 5. Conclusions

In this study, we demonstrated that several somatodendritic ion channels expressed by SNc DA neurons that operate in a coordinated manner to generate a regular and robust autonomous tonic firing appear to be localized in the same macromolecular entity. Specifically, we identified biochemical interactions between Cav1.3, SK3, HCN2, and HCN4 channels. Ion channel tight proximity would be a molecular basis allowing for fine spatiotemporal control of pacemaking activity. Subtle disorganizations of the assembly of this complex could, in the long term, participate in a dysregulation of cellular excitability and rhythmicity and potentially be involved in Parkinson’s Disease symptoms. In addition to the targeted ion channels, we also identified some unknown ion channel interactants, such as Sarm1, strongly associated with HCN4 channels, and Tsc1 and Tsc2, associated with SK3 channels. As it is known that axonal degeneration of dopaminergic neurons is observed at an early stage of the disease and involved in its progression [103], the unexpected presence of Sarm1, an axonal pro-degenerative protein in close proximity with HCN channels in SNc DA neurons, could be of potential interest in understanding the selective vulnerability of these neurons in Parkinson’s Disease. In fact, Sarm1 is involved in induced Parkinson’s-Disease-like pathology [104,105,106,107] and binds to PTEN-induced putative kinase 1 (PINK1) [108], and its activity is L-type-Ca^2+^-channel-dependent [109]. Remarkably, Tsc proteins are also linked to calcium signaling, as Cav1.3 [62] and SK [110] channels were identified as critical downstream components of Tsc-mTOR signaling. Then, the molecular links between the hyperpolarization-activated cyclic nucleotide-gated HCN channels, the L-type calcium channel Cav1.3, and the calcium-activated small-conductance SK3 potassium channels identified in midbrain neurons and, particularly, in SNc DA neurons could help to further understand the underlying factors contributing to the high vulnerability of these neurons and to elaborate novel therapies to slow SNc DA neurodegeneration. This report only constitutes a first step in studying ion channel interactions at the molecular level in SNc DA neurons. Further experiments will be needed to characterize direct interactions in this ion channel assembly. Cross-linking coupled with mass spectrometry (XL-MS) based on the covalent binding of proximal amino acid residues of protein partners [111,112] will help to obtain a fine description of the complex topography of these protein interactions. This approach could be complemented with native gel electrophoresis analysis [113,114]. Then, it would be possible to identify a strategy to dissociate the complex in order to determine its influence on SNc DA neuron pacemaking activity and the vulnerability of this neuronal type in pathological conditions.

## Figures and Tables

**Figure 1 cells-13-00944-f001:**
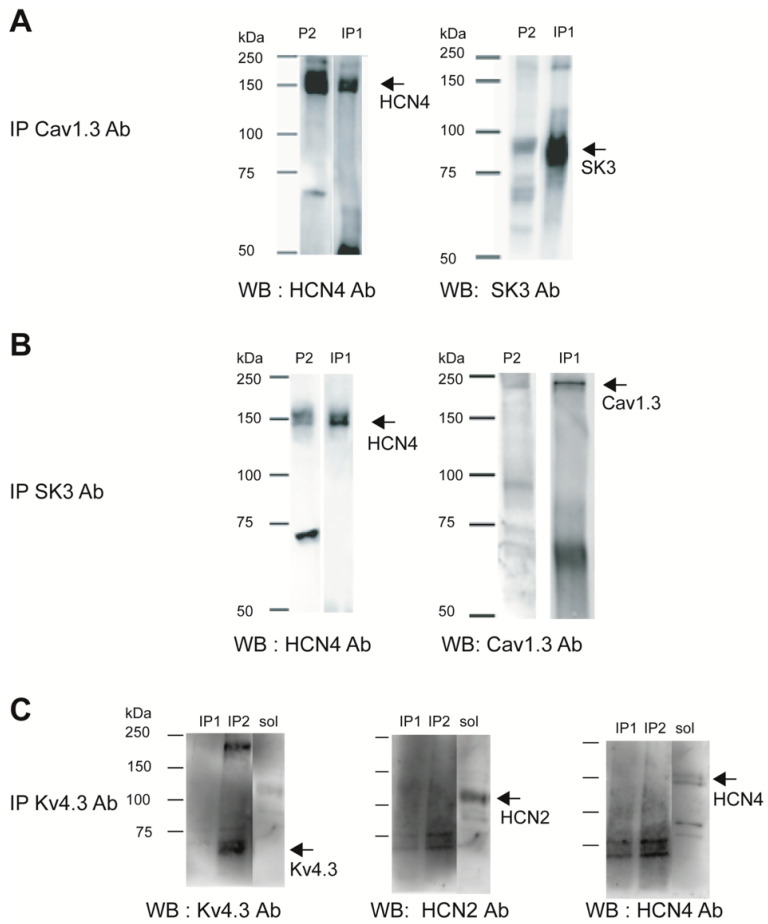
Characterization of the molecular interactions between Cav1.3, SK3, and HCN channels using co-immunoprecipitation and immunoblot. Proteins were membrane-extracted in CL47a solubilization buffer (IP1) or in CL48 buffer (IP2) and immunoprecipitated using anti-Cav1.3 (**A**), anti-SK3 (**B**), and anti-Kv4.3 antibodies (**C**). Blots were performed on starting material, P2 membrane fraction (P2), or on solubilized P2 proteins (sol) or after immunoprecipitation (IP1, IP2). Blots were probed with anti-SK3 ((**A**), right panel), anti-Cav1.3 ((**B**), right panel), anti-HCN4 ((**A**,**B**) left panels; (**C**), right panel), anti-Kv4.3 ((**C**), left panel), and anti-HCN2 ((**C**), middle panel). P2: enriched P2 membrane fraction. IP1: immunoprecipitated proteins in CL47a buffer. IP2: immunoprecipitated proteins in CL48 buffer. sol: solubilisate. WB: Western blot (immunoblot).

**Figure 2 cells-13-00944-f002:**
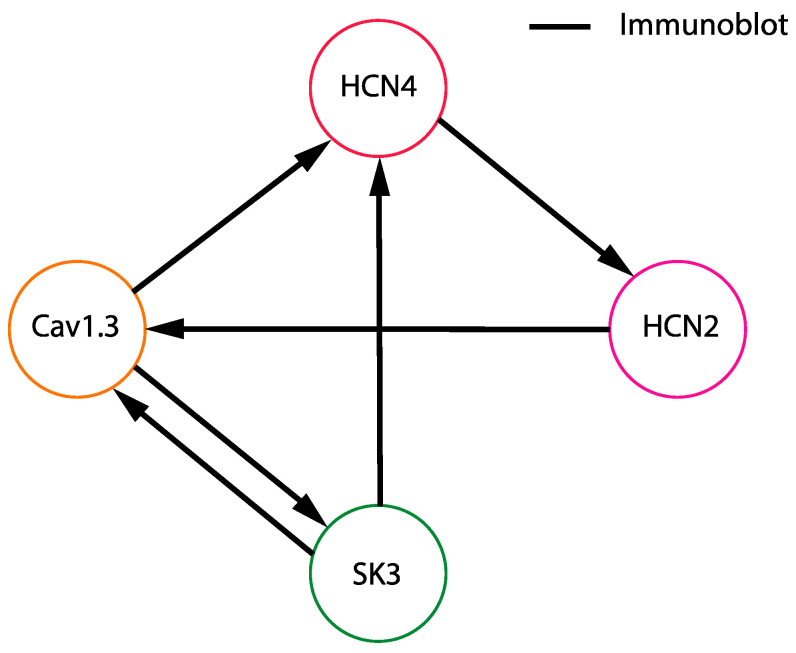
Schematic representation of midbrain ion channel interactions detected through co-immunoprecipitation and immunoblot. Lines indicate co-immunoprecipitated ion channels. The arrows point from the bait to the prey revealed through immunoblot.

**Figure 3 cells-13-00944-f003:**
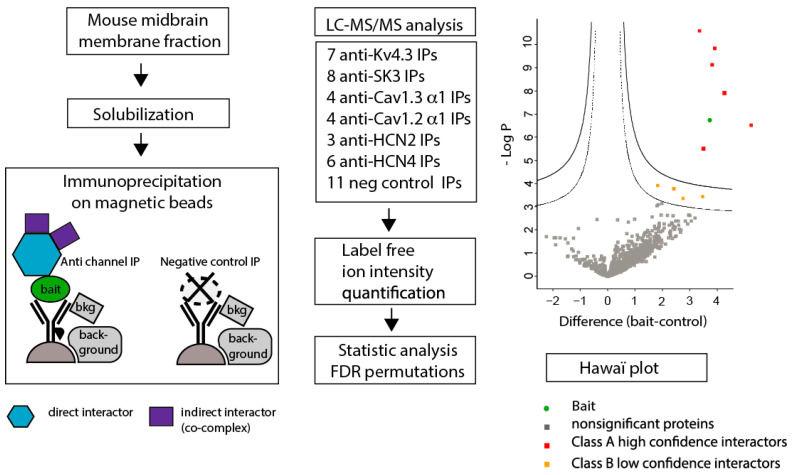
Affinity-purification mass spectrometry protocol design. Plasma-membrane-enriched protein fractions were prepared from acutely dissected midbrain from mice. Immunoprecipitations on wild-type mice were tested against common control consisting of 11 IPs in knock-out mice lacking antigens recognized by used antibodies. Significant interactants are determined using a permutation-based FDR, and the resulting high-confidence Class A (solid line) and low-confidence Class B (dashed lines) thresholds are displayed on the “Hawaii” plot. “Hawaii” plots were used to provide an overview of entire data sets consisting of multiple midbrain membrane immunoprecipitations (IPs).

**Figure 4 cells-13-00944-f004:**
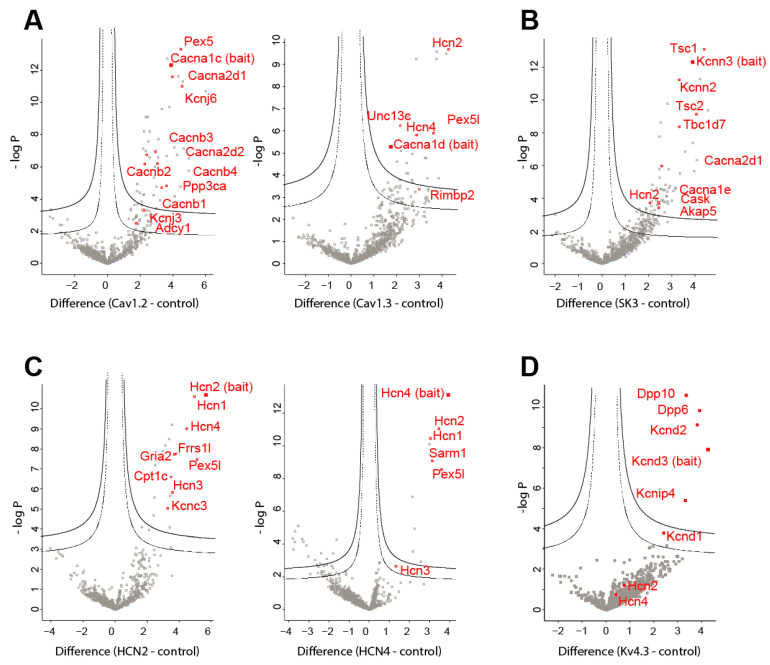
Protein nano-environment of midbrain ion channels determined using affinity-purification mass spectrometry. (**A**) Left panel, Cav1.2 subunit (*Cacna1c* gene) immunoprecipitations. Right panel, Cav1.3 subunit (*Cacna1d* gene) immunoprecipitation. (**B**) SK3 subunit (*Kcnn3* gene) immunoprecipitation. (**C**) Left panel, HCN2 subunit immunoprecipitation. Right panel, HCN4 subunit immunoprecipitation. (**D**) Kv4.3 (*Kcnd3* gene) immunoprecipitation. High-confidence Class A (solid line) and low-confidence Class B (dashed lines) thresholds are displayed on the plots. Plain squares represent the bait and empty square interactants. The most relevant of the significant interactants are named.

**Figure 5 cells-13-00944-f005:**
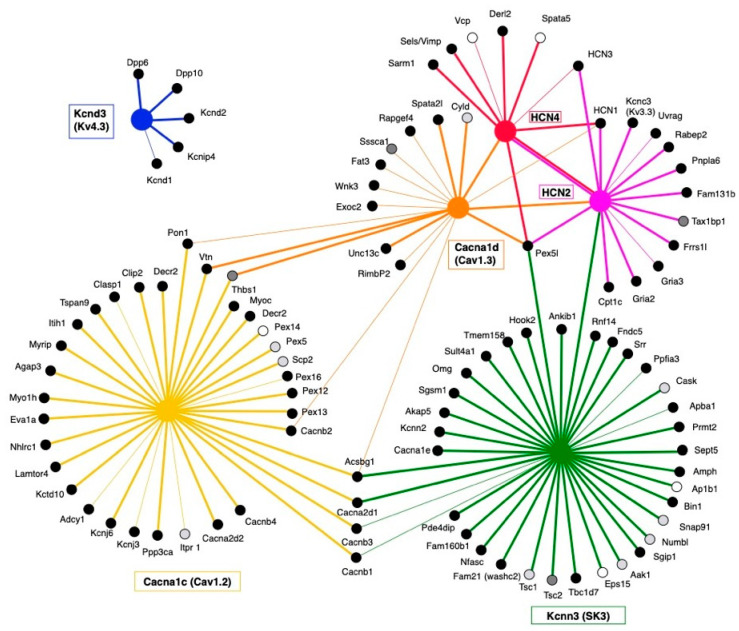
Visualization of the mass-spectrometry-based protein interaction network. The 6 ion channels of interest are depicted using different colors (blue, Kv4.3; pink and red, HCN2 and HCN4; yellow and orange, Cav1.2 and Cav1.3; green, SK3). Interactions are shown with lines of according colors. Thick lines represent Class A interactions, thinner lines Class B. Putative interactants were compared with the CRAPome 2.0 database (Contaminant Repository for Affinity Purification). Proteins identified in less than 1% of all CRAPome negative experiments were considered highly specific (black circles), between 1 and 2%, specific (grey circles), and between 2 and 10%, weakly specific (white circles) interactants.

**Figure 6 cells-13-00944-f006:**
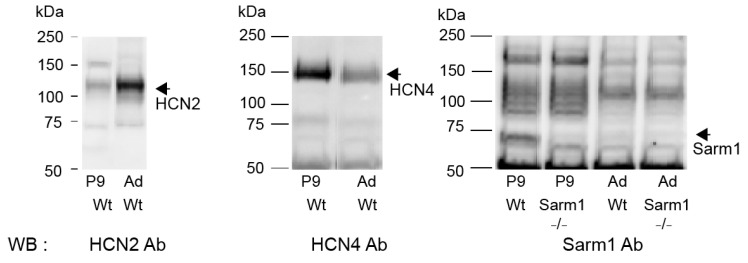
HCN2, HCN4, and Sarm1 expression during midbrain postnatal development. Immunoblots of membrane-enriched fraction from adult (Ad) and 9 (P9) or 11 (P11) postnatal days Wt and Sarm1^−/−^ mice. Blots were probed with anti-HCN2 (left panel), anti-HCN4 (middle panel), or anti-Sarm1 (right panel) antibodies. WB: Western blot (immunoblot).

**Figure 7 cells-13-00944-f007:**
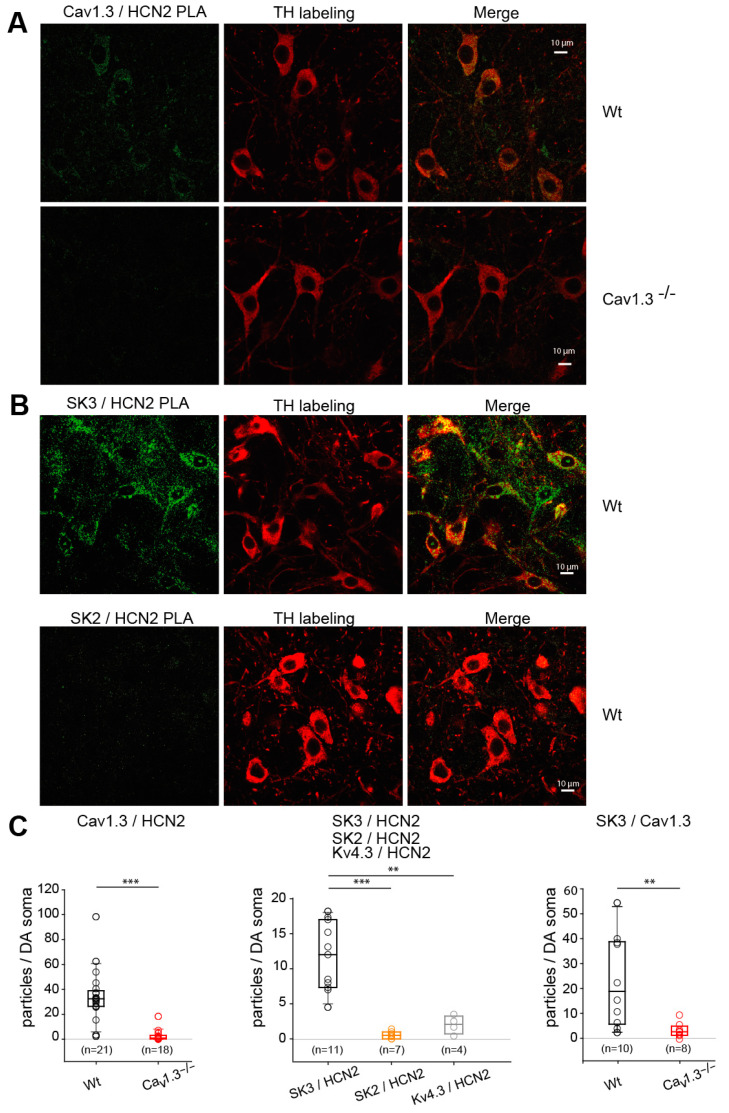
Cav1.3-HCN2-SK3 complex characterization in SNc DA neurons using proximity ligation assay. (**A**) Left, representative images of the PLA reaction (green) in wild-type (Wt) or Cav1.3^−/−^ SNc DA neurons using anti-Cav1.3 and anti-HCN2 antibodies. Middle, immunolabeling of SNc DA neurons with anti-TH antibodies (red). Right, merged images. Scale bars, 10 μm. (**B**) Representative images of the PLA reaction (green) in wild-type (Wt) SNc DA neurons using anti-SK3 and anti-HCN2 antibodies (top row) or anti-SK2 and anti-HCN2 antibodies (bottom row). Middle, immunolabeling of SNc DA neurons with anti-TH antibodies (red). Right, merged images. Scale bars, 10 μm. (**C**) Quantification of PLA fluorescent puncta on SNc DA neurons. Left, Cav1.3 and HCN2 PLA from 3 Wt (empty black circles; *n* = 21 images) or 3 Cav1.3^−/−^ (empty red circles; *n* = 18 images) mice. Superimposed box and whiskers plots allow us to visualize the median and interquartile range. Middle, HCN2 and SK3 PLA (empty black circles; *n* = 11 images) or HCN2 and SK2 (empty orange circles; *n* = 7 images) from 2 independent experiments or anti-Kv4.3 and HCN2 (empty grey circle; *n* = 4 images) PLA from 1 experiment. Superimposed box and whiskers plots allow us to visualize the median and interquartile range. Right, Cav1.3 and SK3 PLA from 2 Wt (empty black circles; *n* = 10 images) or 2 Cav1.3^−/−^ (empty red circles; *n* = 8 images) mice. Superimposed box and whiskers plots allow us to visualize the median and interquartile range. ** *p* < 0.005; *** *p* < 0.001, Wilcoxon–Mann–Whitney test.

**Figure 8 cells-13-00944-f008:**
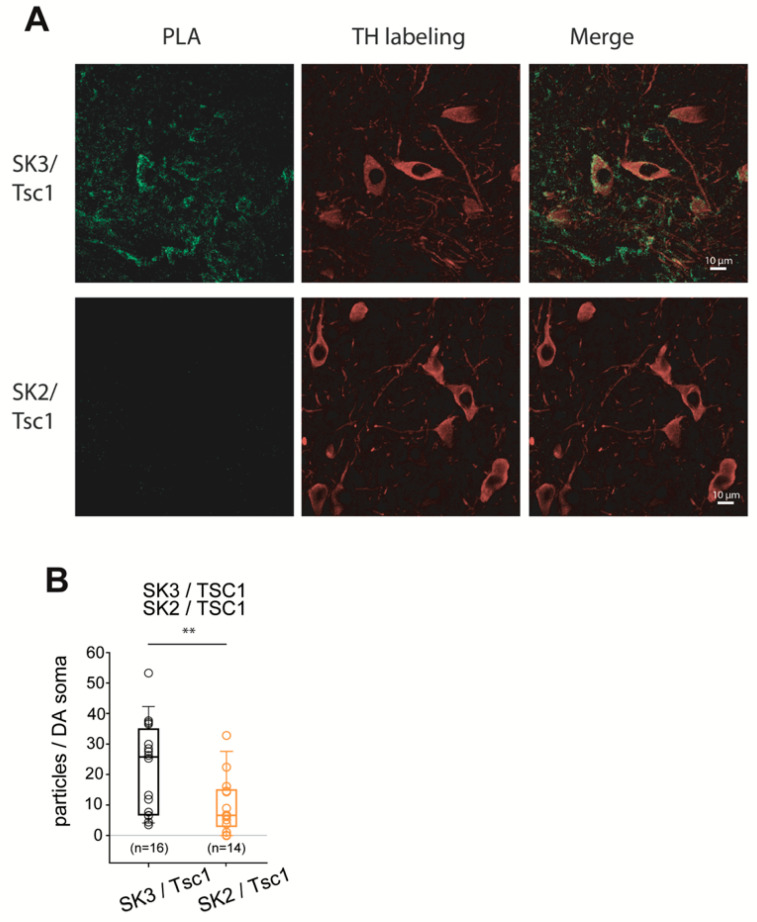
Characterization of the interactions between SK3 and Tsc1 in SNc DA neurons using proximity ligation assay. (**A**) Left, representative images of the PLA reaction (green) in SNc DA neurons using anti-Tsc1 and anti-SK3 antibodies (top row) or anti-Tsc1 and anti-SK2 antibodies (bottom row). Middle, immunolabeling of SNc DA neurons with anti-TH antibodies (red). Right, merged images. Scale bars, 10 μm. (**B**) Quantification of PLA fluorescent puncta in SNc DA neurons using anti-Tsc1 and SK3 antibodies (empty black circles; *n* = 16 images) or anti-Tsc1 and SK2 antibodies (empty orange circles; *n* = 14 images) from 2 independent experiments. Superimposed box and whiskers plots allow us to visualize the median and interquartile range. ** *p* < 0.005, Wilcoxon–Mann–Whitney test.

**Figure 9 cells-13-00944-f009:**
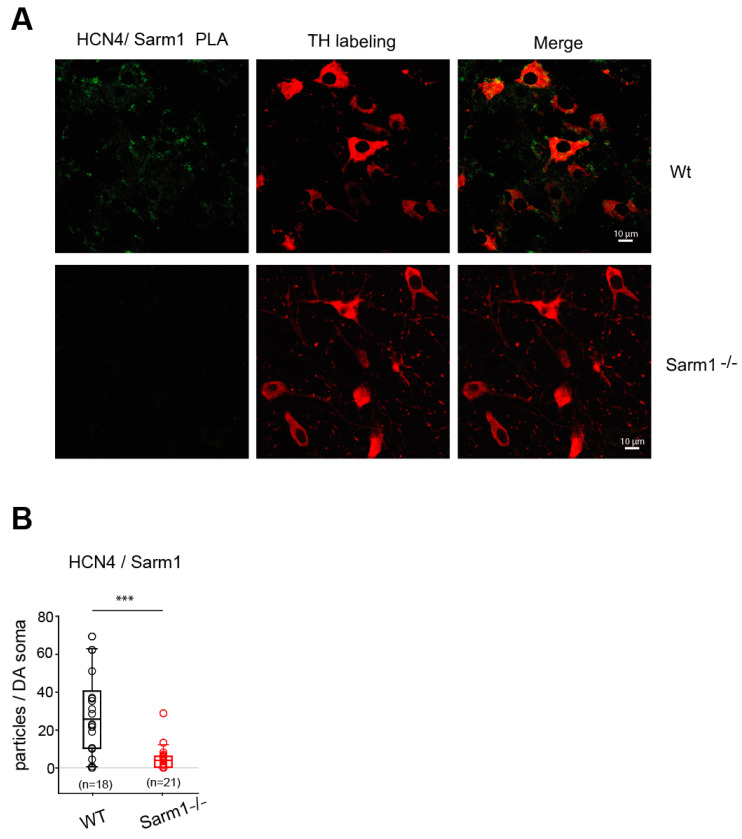
Characterization of the interaction between HCN4 and Sarm1 in SNc DA neurons using proximity ligation assay. (**A**) Left, representative images of the PLA reaction (green) in wild-type (Wt) or Sarm1^−/−-^ SNc DA neurons using anti-Sarm1 and anti-HCN4 antibodies. Middle, immunolabeling of SNc DA neurons with anti-TH antibodies (red). Right, merged images. Scale bars, 10 μm. (**B**) Quantification of PLA fluorescent puncta in SNc slices using anti-Sarm1 and anti-HCN4 antibodies from 2 Wt (empty black circles; *n* = 18 images) or 2 Sarm1^−/−^ (empty red circles; *n* = 21 images) mice. Superimposed box and whiskers plots allow us to visualize the median and interquartile range. *** *p* < 0.001, Wilcoxon–Mann–Whitney test.

**Figure 10 cells-13-00944-f010:**
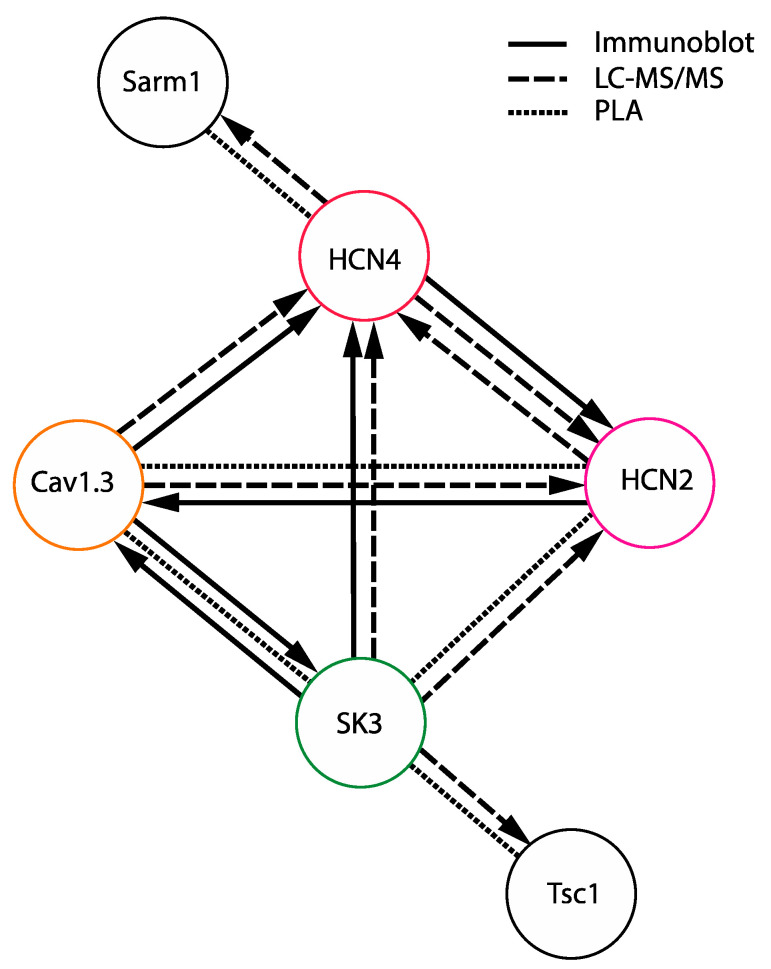
Schematic representation summarizing the ion channel interactions detected with co-immunoprecipitation, mass spectrometry and proximity ligation assay. Plain lines indicate co-immunoprecipitated ion channels. Dashed lines indicate interactions detected in mass spectrometry experiments. The arrows point from the bait to the prey. Dotted lines link proteins detected in close proximity using PLA in situ in SNc DA neurons.

**Table 1 cells-13-00944-t001:** Primary antibodies.

Experiment	Antigen	Host	Type	Companies	Reference	RRID	Concentration ug/mL
Prot A/G	Ca_v_1.3	Rabbit	polyclonal	Alomone	ACC-005	AB_2039775	7
Prot A/G	K_v_43	Mouse	monoclonal	Antibodies Incorporated	75-017	AB_2131966	12
Prot A/G	SK3	Rabbit	polyclonal	Alomone	APC-025	AB_2040130	7
IB	Ca_v_1.3	Rabbit	polyclonal	Alomone	ACC-005	AB_2039775	1.6
IB	HCN2	Rabbit	polyclonal	Alomone	APC-030	AB_2313726	0.85
IB	HCN4	Mouse	monoclonal	Antibodies Incorporated	75-150	AB_2248534	1
IB	HCN4	Rabbit	polyclonal	Alomone	APC-052	AB_2039906	1.3
IB	K_v_4.3	Rabbit	polyclonal	Alomone	APC-017	AB_2040178	1.6
IB	Sarm1	Rabbit	polyclonal	Novus Bio	NBP1-77200	AB_11038887	1
IB	SK3	Rabbit	polyclonal	Alomone	APC-025	AB_2040130	1.6
MS	Ca_v_1.2	Rabbit	polyclonal	Alomone	ACC-003	AB_2039771	ND
MS	Ca_v_1.3	Rabbit	polyclonal	Alomone	ACC-005	AB_2039775	ND
MS	HCN2	Rabbit	polyclonal	Alomone	APC-030	AB_2313726	ND
MS	HCN4	Rabbit	polyclonal	Alomone	APC-052	AB_2039906	ND
MS	K_v_4.3	Rabbit	polyclonal	Alomone	APC-017	AB_2040178	ND
MS	SK3	Rabbit	polyclonal	Alomone	APC-025	AB_2040130	ND
PLA-1	Ca_v_1.3	Mouse	monoclonal	Antibodies Incorporated	75-080	AB_10673964	10
PLA-2	HCN2	Rabbit	polyclonal	Alomone	APC-030	AB_2313726	4
PLA-1	HCN2	Mouse	monoclonal	Antibodies Incorporated	75-111	AB_2279449	10
PLA-2	SK2	Rabbit	polyclonal	Alomone	APC-028	AB_2040126	4
PLA-1	HCN2	Mouse	monoclonal	Antibodies Incorporated	75-111	AB_2279449	10
PLA-2	SK3	Rabbit	polyclonal	Alomone	APC-025	AB_2040130	4
PLA-1	HCN2	Mouse	monoclonal	Antibodies Incorporated	75-111	AB_2279449	10
PLA-2	K_v_4.3	Rabbit	polyclonal	Alomone	APC-017	AB_2040178	2
PLA-1	Ca_v_1.3	Mouse	monoclonal	Antibodies Incorporated	75-080	AB_10673964	10
PLA-2	SK3	Rabbit	polyclonal	Alomone	APC-025	AB_2040130	4
PLA-1	Tsc1	Mouse	monoclonal	Thermo Fisher	37-0400	AB_2533292	2.5; 5
PLA-2	SK3	Rabbit	polyclonal	Alomone	APC-025	AB_2040130	8; 4
PLA-1	Tsc1	Mouse	monoclonal	Thermo Fisher	37-0400	AB_2533292	2.5; 5
PLA-2	SK2	Rabbit	polyclonal	Alomone	APC-028	AB_2040126	8; 4
PLA-1	Sarm1	Mouse	monoclonal	Novus Bio	NBP1-39550	AB_2183996	10
PLA-2	HCN4	Rabbit	polyclonal	Alomone	APC-052	AB_2039906	4

Prot A/G: Protein A/G Sepharose immunoprecipitation; IB: Immunoblot; MS: Mass spectrometry; ND: non determined; PLA-1: Proximity Ligation Assay-1st antibody; PLA-2: Proximity Ligation Assay-2nd antibody.

## Data Availability

The mass spectrometry proteomics data have been deposited to the ProteomeXchange consortium via the Pride Proteomics Identification Database partner repository with the data set identifier PXD-#034883.

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
