# Peer review of "High-Resolution Proteomics Unravel a Native Functional Complex of Cav1.3, SK3, and Hyperpolarization-Activated Cyclic Nucleotide-Gated Channels in Midbrain Dopaminergic Neurons"

_cells, 2024, doi:10.3390/cells13110944_

Round 1

Reviewer 1 Report

Comments and Suggestions for Authors

In the presented work, the authors performed a series of immuno-affinity (CoIP) isolations targeting a selected set of ion channels in midbrain neurons. In a first round of CoIP experiments, obtained protein eluates were analyzed by Western Blotting to demonstrate the presence of the respective target and additional ion channels. Then, they performed larger series of CoIPs under different detergent conditions and with different controls (one target knockout and seven independent antibodies), all in replicates, read-out by high-resolution LC mass spectrometry. Results from each target dataset were then evaluated in volcano plots. These analyses confirmed co-purification of known interactors (mainly auxiliary and heteromeric channel subunits) and provided evidence for new interactors (mainly for CaV 1.2/1.3, SK3 and HCN2) including channel-channel connections. One of latter, the combination Cav1.3-SK3-HCN2, was then probed in midbrain dopaminergic neurons using a proximity ligation assay to claim the existence of a ternary complex of these channels.

Unfortunately, this conclusion was neither further corroborated (e.g. by heterologous reconstitution, native gel analysis) nor investigated functionally, leaving the significance of these findings open.

In addition, I am concerned that - despite the described efforts - antibody specificity may still be an issue. "negative controls are largely 343 bait-independent" is a pretty bold statement, and although CoIPs with target-unrelated antibodies can effectively control for general background proteins, cross-reactivities of individual antibodies will not be removed. This would require either the comparison with CoIPs with additional target-specific antibodies or CoIPs from target knockout tissue (largely missing in this study). The same concern applies to the PLA assays shown (which mostly depended on the same antibodies).

I therefore consider the presented data as too preliminary for publication.

Minor points:

- Volcano plots throughout the ms: x axis has no description. What is the unit of the scale? Linear fold change or log or normalized? Relative enrichment versus control seems quite low even for the target proteins (considered that antibodies should enrich by factors of 1000 and higher).

- Specificity controls: it is not clear from Methods how control data were used. Were the abundances in the 11 control datasets pooled or averaged? Was the same control "pool" used for all CoIPs?

- The rationale underlying data analysis is not entirely clear. Why normalize to the median abundance? How to determine "valid" (versus non-valid) values? Why imputation? Abundance ratios in CoIPs versus control should provide straight-forward information on specificity.

- In addition to specificity, the co-purification efficiency (comparison of abundance of co-purified proteins compared to target) should be determined and evaluated.

- Fig. 5. "crap-ome" is not a meaningful resource for determination of significance (in particular when cross reactivities are not ruled out).

- Please check labelling in figures for spelling errors (e.g. "controle" instead of "control", "Frrsl1" instead of Frrs1l)

- Supplementary information should be provided on protein MS identification (peptides/matches) and quantification (mol. abundance of protein) in each measurement performed.

- Please specify the "irrelevant antibodies" used. Why were these controls made with Kv4.3 ko rather than with WT membranes?

- A number of detergent conditions were tested but only results for two of them shown. This selection and potential (in)consistencies with other conditions should at least be discussed.

- MS Methods: "match between runs" is a quite relevant feature for robust label-free quantification. When measurements were indeed "too much spread over time" datasets should be either separately evaluated (with "match between runs") or repeated under the same conditions.

Comments on the Quality of English Language

Many spelling errors in text and figures.

Author Response

Review 1-March 18, 2024

Comments and Suggestions for Authors

In the presented work, the authors performed a series of immuno-affinity (CoIP) isolations targeting a selected set of ion channels in midbrain neurons. In a first round of CoIP experiments, obtained protein eluates were analyzed by Western Blotting to demonstrate the presence of the respective target and additional ion channels. Then, they performed larger series of CoIPs under different detergent conditions and with different controls (one target knockout and seven independent antibodies), all in replicates, read-out by high-resolution LC mass spectrometry. Results from each target dataset were then evaluated in volcano plots. These analyses confirmed co-purification of known interactors (mainly auxiliary and heteromeric channel subunits) and provided evidence for new interactors (mainly for CaV 1.2/1.3, SK3 and HCN2) including channel-channel connections. One of latter, the combination Cav1.3-SK3-HCN2, was then probed in midbrain dopaminergic neurons using a proximity ligation assay to claim the existence of a ternary complex of these channels.

Unfortunately, this conclusion was neither further corroborated (e.g. by heterologous reconstitution, native gel analysis) nor investigated functionally, leaving the significance of these findings open.

In addition, I am concerned that - despite the described efforts, antibody specificity may still be an issue. "negative controls are largely 343 bait-independent" is a pretty bold statement, and although CoIPs with target-unrelated antibodies can effectively control for general background proteins, cross-reactivities of individual antibodies will not be removed. This would require either the comparison with CoIPs with additional target-specific antibodies or CoIPs from target knockout tissue (largely missing in this study). The same concern applies to the PLA assays shown (which mostly depended on the same antibodies).

I therefore consider the presented data as too preliminary for publication.

Unfortunately, this conclusion was neither further corroborated (e.g. by heterologous reconstitution, native gel analysis) nor investigated functionally, leaving the significance of these findings open.

Reply

Our study constitutes a first step in the identification of a nano-environment of ion channels known to functionally interact in SNc DA neurons. It is based on a careful analysis of a large number of immuno-purifications of poorly soluble and low-abundance ion channels. We do not believe and did not claim that interactions between these ion channels are direct. If interactions are mostly indirect, reconstituting the complex in a heterologous expression system would be impossible. Even in the opposite case, ion channel structure reconstitution and addressing to the plasma membrane are quite challenging and these experiments would be particularly difficult to achieve. The next step of our study will rather be to decipher the identified complex by Cross-linking coupled to Mass Spectrometry (XL-MS) based on the covalent binding of proximal amino acid residues of protein partners with a cross-linker chemical reagent. Native gel analysis of IPs would not bring further proof of specificity on the ion channel assembly, but will give access to their native abundance and stoichiometries. These experiments will complement the future experiments suggested above. Then, it will be possible to identify a strategy to dissociate the complex and determine the influence of this manipulation on SNc DA neuron pacemaking activity and vulnerability. To answer the reviewer’s concern, we modified the conclusion section, line 696:

“This report only constitutes a first step in studying ion channel interactions at the molecular level in SNc DA neurons. Further experiments will be needed to characterize direct interactions in this ion channel assembly. Cross-linking coupled to mass spectrometry (XL-MS) based on the covalent binding of proximal amino acid residues of protein partners will help to obtain a fine description of the complex topography of these protein interactions. This approach could be complemented with native gel electrophoresis analysis. Then it would be possible to identify a strategy to dissociate the complex in order to determine its influence on SNc DA neuron pacemaking activity and on vulnerability of this neuronal type in pathological conditions.”

In addition, I am concerned that – despite the described efforts – antibody specificity may still be an issue. “negative controls are largely bait-independent” is a pretty bold statement, and although CoIPs with target-unrelated antibodies can effectively control for general background proteins, cross-reactivities of individual antibodies will not be removed. This would require either the comparison with CoIPs with additional target-specific antibodies or CoIPs from target knockout tissue (largely missing in this study). The same concern applies to the PLA assays shown (which mostly depended on the same antibodies).

Reply

We thank reviewer 1 requesting us to present in a clearer way our strategy to detect specific interactor in our experiments. See below our answers to his main concerns.

Concerning specificity of MS/MS data

We took extremely care to minimize false-positive interactor without extra filtering specific interactor using a strategy approved by the research community [1] to remove as much as possible of contaminants from the sample in MS:MS experiments.

Biochemical design before MS/MS

In our biochemical experimental design, we chose to use magnetic beads as a support of immunoprecipitation. Their reduced size, hydrophobicity and porosity have an effective positive impact on purification specificity.

We performed a short incubation of antibodies for only 2 hours since many studies have demonstrated that rapid purifications ensure specific interactor enrichment [2, 3, 4].

Antibodies

Antibodies against ion channels from Alomone are recognized as highly specific and largely used by the scientific community. Anti- Cav1.3 antibody has undergone quality control on Cav1.3-/- KO and is cited in 186 articles.

Controls

Negative controls: we took much attention to incorporate many proper negative controls. We thus performed experiments on target knock-out tissues being the most meaningful ones. As most of our targets are ion channels essential for brain and also heart functions, it was not possible to produce KO mice for each chosen target. We performed 4 independent IPs using anti-Kv4.3 on Kv4.3-/- KO mice. We also raised Cav1.3-/- KO mice but due to the poor health of these animals and low detection of Cav1.3 channels we could perform too few MS/MS experiments. But we included 7 IPs using antibody against Sarm1 on Sarm1-/- knock out. Although Sarm1 was not one of the targeted ion-channel, most unspecific interactors proteins were assigned in these negative controls.

Our large study using in parallel seven antibodies targeting different baits led us to discriminate unspecific binding as different antibodies are less likely to cross-react with the same background proteins.

Biological replicates

Many biological replicates were done for each of the 7 targets for a total of 43 independent immuno-purifications, a key for the identification of robust interactions.

Quantification of MS/MS signal

Multiple research groups noticed that the best way to characterize specific interaction using MS/MS is to monitor quantitative enrichment in the mass spectrometer. Therefore, we measured the intensity of fragment ions corresponding to a given interactor (the prey) in the purification of the bait and compared it to the “noise” corresponding to the same prey in the control analyses. This gives the possibility to measure enrichment in the mass spectrometer and to discriminate false-positive data as a true interactor is enriched throughout purification.

Statistical approach

Data statistic comparisons of all the IPs experiments with a probabilistic method using Perseus gave high confidence to detect significant interactors.

Contaminant repository data

The use of a contaminant repository data for affinity purification (CRAPome) made of negative controls generated by the proteomics research community allowed the capture of the most complete set of contaminants. That strategy is based on the shared observations through large data samples that negative controls are largely bait independent.

Known interactors

As an internal validation, we identified several ion channel paralogs as well as auxiliary subunits and already published interactors of the ion channels included in our study, thus confirming the efficiency of our strategy.

We are replacing the paragraph 3.2, line 335 by the paragraph below, explaining with more details the design of the AP-MS experiment, stressing on our concern to minimize detection of false-positive interactors while not eliminating specific binding partners.

“In order to minimize false-positive identification of background proteins while not extra filtering proteins specifically recognized by antibodies, we used magnetic beads as a support of immunoprecipitation since their reduced size, hydrophobicity and porosity have an effective positive impact on purification specificity [1]. Also, short antibody incubations of only 2 hours were used since many studies have demonstrated that rapid purifications ensure specific interactant enrichment [2], [3], [4].

We also incorporated many proper negative controls. We thus performed experiments on target knock-out tissues whenever it was possible. As most of our targets are ion channels essential for brain and heart functions, it was not possible to produce KO mice for each chosen target. We performed 4 independent IPs using anti-Kv4.3 antibodies on Kv4.3-/- KO mice. We also raised Cav1.3-/- KO mice but due to the poor health of these animals and low detection of Cav1.3 channels we could not perform MS/MS experiments. So, we included 7 IPs using antibody against an unrelated target, Sarm1, on Sarm1-/- KO membranes. Although Sarm1 was not one of the targeted ion channels, most non-specific interacting proteins were assigned in these negative controls.

Our large study using in parallel seven antibodies targeting different baits led us to discriminate unspecific binding, as different antibodies are less likely to cross-react with the same background proteins. Furthermore, many biological replicates of immunoprecipitations (IPs), key to the identification of robust interactions, were done on wild-type mice based on 4, 4, 3, 6, 7 and 8 replicates of Cav1.2, Cav1.3, HCN2, HCN4, Kv4.3, and SK3 subunits, respectively, for a total of 43 independent immuno-purifications.

Multiple research groups noticed that the best way to characterize specific interactions using MS/MS is to monitor quantitative enrichment in the mass spectrometer. Therefore, we measured the intensity of peptide ions corresponding to a given interactant (the prey) in the purification of the bait and compared it to the “noise” corresponding to the same prey in the control analyses. This gives the possibility to measure enrichment in the mass spectrometer and to discriminate false-positive data, as true interactants are enriched throughout purification. Statistical comparisons of all the data from IP experiments with a probabilistic method using Perseus software gave high confidence to detect significant interactants.

Finally, the use of a contaminant repository data for affinity purification (CRAPome) made of negative controls generated by the proteomics research community allowed the capture of the most complete set of contaminants. That strategy is based on the shared observations through large data samples that negative controls are largely bait-independent [5].

As an internal validation, we identified several ion channel isoforms as well as auxiliary subunits and already published interactants of the ion channels included in our study, thus further confirming the efficiency of our strategy.”

PLA experiments

MS/MS data were validated by PLA experiments. Cross reactivity of anti-Cav1.3 and anti-SK3 antibodies toward HCN2 proteins, while being very unlikely, cannot account for the results showing close proximity of Cav1.3 and SK3 in wild-type tissue and not in Cav1.3-/- samples, further pointing that the three ion channels are in the same nano-environment. So, we added in the manuscript, line 492:

“Also, SK3 subunits were found in close proximity with HCN2 proteins (Figure 7B, top row and C middle panel) but not SK2 (Figure 7B, bottom row and C middle panel), nor Kv4.3 (Figure 7C, middle panel) subunits. PLA analysis revealed that Cav1.3 and SK3 proteins were also in close proximity in SNc DA neurons (Figure 7C right panel), strengthening the observation that HCN2 interactions depicted using anti-Cav1.3 and anti-SK3 channel antibodies were not due to antibody cross-reactivity.”

Minor points:

- Volcano plots throughout the ms: x axis has no description. What is the unit of the scale? Linear fold change or log or normalized? Relative enrichment versus control seems quite low even for the target proteins (considered that antibodies should enrich by factors of 1000 and higher).

The protein abundances are log2-transformed and then normalized using Z-score. Perseus volcano plots represent differential protein abundance between the tested sample (bait) and control groups.

The unit of the x axis was indicated as “bait - control” (read bait minus control) which was not clear, so we changed it to “difference (bait – control)”on Figure 4. This is equivalent to log2(bait/ control).

Adjustments of data point by normalization change the scale of the data and consequently affect the interpretation of fold changes. After normalization, the general trends and patterns in the data are still informative for identifying differentially abundant proteins between conditions, but it does not correspond to absolute differences in abundance level.

- Specificity controls: it is not clear from Methods how control data were used. Were the abundances in the 11 control datasets pooled or averaged? Was the same control "pool" used for all CoIPs?

The control dataset of the 11 negative control immunoprecipitations were not averaged but used as replicate controls for statistical analysis. The same 11 negative control experiments were used for all the relevant immunoprecipitations.

- The rationale underlying data analysis is not entirely clear. Why normalize to the median abundance? How to determine "valid" (versus non-valid) values? Why imputation? Abundance ratios in CoIPs versus control should provide straight-forward information on specificity.

We used a workflow analysis, adapted from [6], to pre-process the data and make them ready for statistical analysis. This includes:

  • log2 transformation of protein intensities
  • Normalization
  • Filtering of valid values (a log2(0) is a non-valid value).
  • Imputation of remaining non-valid values (missing values) with low-range values in the normal distribution.

Normalization, filtering, and imputation are recommended practices to reach statistically meaningful conclusions. Normalizing with the Z-score enables to compare two different scores from different normal distributions of the data. The use of the median is useful since it is less sensitive to outliers than the mean. We thus added line 236:

“MaxQuant search results were exported to Perseus platform (version 1.6.15) for statistical analysis. The data transformation was adapted from the general proteomics workflow described in [6]. Protein quantification was based on the intensity-based absolute quantification (iBAQ) algorithm integrated into the MaxQuant platform. Reverse and “only identified by site” identifications were excluded from further data analysis. After log2 transformation of the leftover proteins, iBAQ values were normalized by Z-score in which the median of each sample column is subtracted from each value and then divided by the standard deviation of the column. Normal distribution was then checked graphically. Contaminants were then filtered out of the identified protein list and proteins were kept if at least 66% valid values were present in at least one protein group (control or baits). Missing values were then imputed assuming a normal distribution of 0.3 width and 3 downshifts. Significant data points were determined using a 200-permutation based FDR with a Pearson correlation function using the Hawaii plot tool (multiple volcano plots) implemented in Perseus. Statistical testing using the default parameters allowed to determine positive interactants classified into Class A (1% FDR, S0=0.1) or Class B (5% FDR, S0=0.1) [7].”

- In addition to specificity, the co-purification efficiency (comparison of abundance of co-purified proteins compared to target) should be determined and evaluated.

Adjustments of data point by normalization change the scale of the data and consequently affect the interpretation of fold changes. After normalization, the general trends and patterns in the data are still informative for identifying differentially abundant proteins between conditions, but it does not correspond to absolute differences in abundance level.” So, abundance values of co-purified proteins can be compared to the bait in a given volcano plot.

- Fig. 5. "crap-ome" is not a meaningful resource for determination of significance (in particular when cross reactivities are not ruled out).

Extensive elimination of background contaminants makes it difficult to distinguish between true low-level interactors and nonspecific interactors and, as such, effective methods for the identification of contaminants were developed. Contaminant lists for affinity purification on different supports and from different organisms disseminated throughout the literature and efforts underway across several laboratories to create a large Contaminant Repository of more than 700 Affinity-Purification mass spectrometry data from negative control was a real advancement for the following reasons:

  1. Even if knock-out animals would be available to use in negative controls for all targets, the absence of detection of a protein in one control run performed in parallel to the sample is not sufficient to conclude that this is a specific interactor for the protein of interest due to the variability between each independent purification;
  2. Nonspecific interactors are largely independent of the antibodies used in affinity-purification as observed throughout the literature and recalled by Mellacheruvu [5] presenting the “Contaminant repository for affinity purification–mass spectrometry data”, as many different antibodies are less likely to cross-react with the same background proteins.
  • We did not use the CRAPome data to increase the significance of our identified interactants, nor to score the interactions, but only to eliminate non-specific ones. The only risk is to have “missed” a significant interactant, but we do not claim to have an exhaustive list of all baits interactants.

- Please check labelling in figures for spelling errors (e.g. "controle" instead of "control", "Frrsl1" instead of Frrs1l)

Corrections have been made in figures 3 and 4.

Supplementary information should be provided on protein MS identification (peptides/matches) and quantification (mol. abundance of protein) in each measurement performed.

Table S2 of all peptide sequences is now provided as supplementary information with quantification data of all the identified proteins. 

- Please specify the "irrelevant antibodies" used. Why were these controls made with Kv4.3 ko rather than with WT membranes?

Yes, the use of the expression “irrelevant antibody” was not correct. Negative control experiments were done using either Kv4.3 or Sarm1 antibodies on membrane fractions from mice deleted with the corresponding antigen (Kcnd3-/- and Sarm1-/- mice respectively). The reason was to identify background proteins. In Sarm1 experiments it is more accurate to say that we used as negative control an antibody recognizing irrelevant target knockout tissue. We changed the sentence in the manuscript and detailed the name of the unrelated target, line 347:

“We performed 4 independent IPs using anti-Kv4.3 antibodies on Kv4.3-/- KO mice. We also raised Cav1.3-/- KO mice but due to the poor health of these animals and low detection of Cav1.3 channels we could not perform MS/MS experiments. So, we included 7 IPs using antibody against an unrelated target, Sarm1, on Sarm1-/- KO membranes. Although Sarm1 was not one of the targeted ion channels, most non-specific interacting proteins were assigned in these negative controls.”

 A number of detergent conditions were tested but only results for two of them shown. This selection and potential (in)consistencies with other conditions should at least be discussed.

Using an immunoblot approach, we could establish that comparable amounts of HCN4 were co-purified with Cav1.3 (Fig.1 left panel) and SK3 (Fig. 1B left panel) channels when solubilizing the membranes with 8 Complexiolyte buffers (Logopharm, GmbH) of varying stringencies. Subsequently, we presented results obtained using the CL47a buffer exclusively. Immunoprecipitations of Kv4.3 were also done in the 8 Complexiolyte buffers and did not reveal the presence of HCN2 and HCN4 in any of the samples. In addition to showing the results of immunoprecipitation using the CL47a buffer we presented the results in buffer CL48 (Fig.1C) where the highest amount of Kv4.3 was immunoprecipitated, further strengthening the absence of any interactions with HCN2 and HCN4.

To clarify our conclusions, we removed sentences in Material and Methods line 142 and added sentences in the Results section; lines 298 and 305. We changed in Figures 1A and 1B “IP” for “IP1” illustrating that immunoprecipitations were done in CL47a buffer as in Figure 1C.

142

“Experiments to establish HCN interactions were done in all 8 solubilization buffers. with no difference in the results. Only the results using CL47a buffer in Cav1.3 and SK3 channel immunoprecipitations and CL47 and CL48 in Kv4.3 immunoprecipitations are shown.”

297

“Optimization of extraction and purification conditions were individually searched for the six channel protein complexes testing eight solubilization buffers of varying stringencies from Logopharm (GmbH). The solubilization buffer CL47a able to efficiently extract the six channel subunits from the lipidic membrane was selected to perform co-immunoprecipitation experiments.”

305

“Based on the literature, we focused on the potential interactions between Cav1.3, Kv4.3, HCN2, HCN4 and SK3 channels. Mouse midbrain plasma membrane-enriched protein fractions were used as starting material (Figure 1A, 1B, lane P2). HCN4 channel subunits were co-immunoprecipitated with Cav1.3 (Figure 1A, left panel, lane IP1) and SK3 (Figure 1B, left panel, lane IP1) channel proteins in buffer CL47a. HCN4 co-purifications were found in all solubilization buffers of varying stringencies tested (data not shown), suggesting strong interactions of HCN4 with Cav1.3 and HCN4 with SK3 channels. Furthermore, SK3 proteins were immunoprecipitated with anti-Cav1.3 antibodies (Figure 1A, right panel) and Cav1.3 subunits with anti-SK3 antibodies (Figure 1B, right panel) in the CL47a solubilization buffer. No molecular interactions were observed between Kv4.3 and HCN2 or HCN4 channels (Figure 1C, middle and right panels, respectively). Although HCN2 (Figure 1C, middle panel) and HCN4 (Figure 1C, right panel) channels were observed in the solubilized starting material (sol) before immunoprecipitation, they were not co-purified with Kv4.3 channels immunoprecipitated in buffers CL47a (Figure 1C, lane IP1) or CL48 (Figure 1C, lane IP2), the most powerful buffer for Kv4.3 channel extraction. The same results were obtained in all solubilization buffers of varying stringencies tested (data not shown). However, we cannot exclude any binding of Kv4.3 channels with the identified macromolecular complex as Kv4.3 proteins appear more difficult to extract from the membranes than the other channels (see the absence of Kv4.3 labelling in the solubilized starting material on Figure 1C, left panel, lane sol). Furthermore, Kv4.3-immunolabeled bands of larger apparent molecular weights were observed, indicating possibly the presence of Kv4.3 channels in large partially denaturated complexes.”                                                                     

- MS Methods: "match between runs" is a quite relevant feature for robust label-free quantification. When measurements were indeed "too much spread over time" datasets should be either separately evaluated (with "match between runs") or repeated under the same conditions.

We acknowledge that the 'match between runs' (MBR) option is highly effective, in the case of identical data, in reducing missing values in experiments, thereby enhancing quantification. MBR enables the identification and quantification of parent ions even when MS/MS data for identification are absent, by deducing it through retention time alignment of the same charged state and m/z ion from another run. However, it is important to note that this matching is limited to replicates processed on the same batch of LC-MS/MS runs with small retention time variations. When retention times differ by more than a few minutes or if the samples are different, this option can lead to false identifications. In our study, the samples are not all identical, and even for replicates, the extensive number of immunoprecipitation analyses conducted imposed the use of different batches of chromatographic columns and pre-columns, resulting in non-superimposable chromatograms. Moreover, the publication [8] from Lim et al., (2019) points out a 2.7% transfer of wrong identifications in similar samples. Therefore, we made the decision not to align our chromatograms to prevent misidentification.

The text was modified line 231 to:

Minimum peptide length was set to 7 and “match between runs” was disabled to avoid misidentification [8].

Comments on the Quality of English Language

Many spelling errors in text and figures. Errors were removed.

References

[1]        W. H. Dunham, M. Mullin, et A.-C. Gingras, « Affinity-purification coupled to mass spectrometry: Basic principles and strategies », PROTEOMICS, vol. 12, no 10, p. 1576‑1590, 2012, doi: 10.1002/pmic.201100523.

[2]        I. M. Cristea, R. Williams, B. T. Chait, et M. P. Rout, « Fluorescent proteins as proteomic probes », Mol Cell Proteomics, vol. 4, no 12, p. 1933‑1941, déc. 2005, doi: 10.1074/mcp.M500227-MCP200.

[3]        M. Oeffinger et al., « Comprehensive analysis of diverse ribonucleoprotein complexes », Nat Methods, vol. 4, no 11, p. 951‑956, nov. 2007, doi: 10.1038/nmeth1101.

[4]        C. S. Müller et al., « Quantitative proteomics of the Cav2 channel nano-environments in the mammalian brain », Proc Natl Acad Sci U S A, vol. 107, no 34, p. 14950‑14957, août 2010, doi: 10.1073/pnas.1005940107.

[5]        D. Mellacheruvu et al., « The CRAPome: a contaminant repository for affinity purification–mass spectrometry data », Nat Methods, vol. 10, no 8, Art. no 8, août 2013, doi: 10.1038/nmeth.2557.

[6]        S. Tyanova et J. Cox, « Perseus: A Bioinformatics Platform for Integrative Analysis of Proteomics Data in Cancer Research », Methods Mol Biol, vol. 1711, p. 133‑148, 2018, doi: 10.1007/978-1-4939-7493-1_7.

[7]        J. D. Rudolph et J. Cox, « A Network Module for the Perseus Software for Computational Proteomics Facilitates Proteome Interaction Graph Analysis », J Proteome Res, vol. 18, no 5, p. 2052‑2064, mai 2019, doi: 10.1021/acs.jproteome.8b00927.

[8]        M. Y. Lim, J. A. Paulo, et S. P. Gygi, « Evaluating False Transfer Rates from the Match-between-Runs Algorithm with a Two-Proteome Model », J Proteome Res, vol. 18, no 11, p. 4020‑4026, nov. 2019, doi: 10.1021/acs.jproteome.9b00492.

Reviewer 2 Report

Comments and Suggestions for Authors

The manuscript #cells-2924413 entitled "High-resolution proteomics unravel a native functional complex of Cav1.3, SK3 and HCN channels in midbrain dopaminergic neurons" and co-authored by Belghazi et al. [B. Marquèze-Pouey, corresponding author] deals with the molecular unraveling of a supramolecular complex, constistuted by
native ion channels (i.e., Cav1.3, SK3 and HCN) underlying the pacemaking activity of dopaminergic neurons in the substantia nigra of mice. In addition, molecular interactants of ion channel activity were identified and analyzed.

In my opinion, this is what a scientific article shoud be: novel, well conceived, well conducted, well written and, finally, informative.

The novelty of both rationale and approach resides in the analysis of neuronal excitability at macromolecular level, where not only the activity of a single channel is relevant as well as its network interactions with other channels. This is a new and important approach which will challenge the study of the role of ion channels in the physiological and pathological neuron biology.

The experimental section has been multidisciplinarly conducted and the results are convincing. In general the manuscript is well organized and clearly written, results are well treated, exposed and interpreted (the iconographic set is of good quality).

As a whole, this manuscript is almost flawless and surely worth of publication in Cells.

Just to improve the value of the manuscript I would suggest some text and refs additions whose locationt (introduction and/or discussion) I leave to the Authors' decision.

The study is focussed on molecular/structural study of the macromolecular complex but it will benefit from a functional framework that is is only barely mentioned. In particular:

1) the common localization of macromolecular complex could have a corresponding common functional activity (to this regard I think e.g. to cross-correlation study already ad-oc conducted on vertebrate and invertebrate nervous system).

2) the identification and the role of other ion channel macromolecular complex in the normal physiology of other neuronal districts.

3) the identification and the role of other ion channel macromolecular complex in the phatophysiology of neurological disorders.

Finally, in the Results 3.1. section (line 311), Authors should explain better why Kv4.3 proteins " appear more difficult to extract".

Author Response

Review 2-March 25, 2024

Comments and Suggestions for Authors

The manuscript #cells-2924413 entitled "High-resolution proteomics unravel a native functional complex of Cav1.3, SK3 and HCN channels in midbrain dopaminergic neurons" and co-authored by Belghazi et al. [B. Marquèze-Pouey, corresponding author] deals with the molecular unraveling of a supramolecular complex, constistuted by
native ion channels (i.e., Cav1.3, SK3 and HCN) underlying the pacemaking activity of dopaminergic neurons in the substantia nigra of mice. In addition, molecular interactants of ion channel activity were identified and analyzed.

In my opinion, this is what a scientific article shoud be: novel, well conceived, well conducted, well written and, finally, informative.

The novelty of both rationale and approach resides in the analysis of neuronal excitability at macromolecular level, where not only the activity of a single channel is relevant as well as its network interactions with other channels. This is a new and important approach which will challenge the study of the role of ion channels in the physiological and pathological neuron biology.

The experimental section has been multidisciplinarly conducted and the results are convincing. In general the manuscript is well organized and clearly written, results are well treated, exposed and interpreted (the iconographic set is of good quality).

As a whole, this manuscript is almost flawless and surely worth of publication in Cells.

Just to improve the value of the manuscript I would suggest some text and refs additions whose locationt (introduction and/or discussion) I leave to the Authors' decision.

The study is focused on molecular/structural study of the macromolecular complex but it will benefit from a functional framework that is is only barely mentioned. In particular:

1) the common localization of macromolecular complex could have a corresponding common functional activity (to this regard I think e.g. to cross-correlation study already ad-oc conducted on vertebrate and invertebrate nervous system).

2) the identification and the role of other ion channel macromolecular complex in the normal physiology of other neuronal districts.

3) the identification and the role of other ion channel macromolecular complex in the pathophysiology of neurological disorders.

We thank reviewer 2 to make us improving the value of our publication by refining the introduction and discussion.

Reply to point 1

We better discussed the link between the common localization and function of macromolecular complex by adding a new chapter in the discussion after line 672 called “Putative physiological and physiopathological meaning of the ion channel macromolecular complex”.

“Putative physiological and physiopathological meaning of the ion channel macromolecular complex

As summarized in Figure 10, interactions determined here, depict that several ion channels are organized as a macromolecular complex in SNc DA neurons: Cav1.3, HCN2, HCN4, and SK3. Surprisingly, Kv4.3 does not belong to this complex. As already extensively commented, all these ion channels have several features in common: i) they are all mainly located in the somatodendritic compartment where ii) their role is to control the pace and regularity of the spontaneous activity of SNc DA neurons. The question that arises then is how this molecular organization might affect the functional interactions between these ion channels. As already mentioned, in spite of the co-variation of voltage dependences of IA and IH currents [1], Kv4.3 and HCN channels were not found to interact at the molecular level, meaning that co-regulation by the same signaling pathways may not imply a strict nanometric co-localization of channels. Interestingly, concerning HCN and Cav1.3 ion channels, one study suggested that pacemaking might rely on the synergistic actions of HCN, Cav1.3 and Nav channels, such that, in the absence of Cav1.3, pacemaking might rejuvenate to an HCN/Nav-dependent mechanism [2]. Although we did not investigate the potential interactions between Nav, HCN and Cav1.3 channels in the current study, it is tempting to speculate that the co-localization of these channels may partly underlie their synergistic effects on pacemaking. Concerning Cav1.3 and SK3, several studies investigated the calcium sources responsible for SK channel activation in SNc DA neurons [98, 16, 99]. In two of these studies [16, 99], the influence of L-type calcium (Cav1) channels to activate apamin-sensitive (SK) currents was tested and both studies concluded that Cav1 channels do not support the calcium entry necessary for SK channel activation, although several types of calcium channels (including different subtypes of Cav2 and Cav3 channels) appear to mediate it [98, 16, 99]. Thus, the presence of Cav1.3 and SK3 complex contrasts with the demonstrated lack of dependence of SK3 activation on Cav1-mediated calcium entry. In one of these studies though [5], Cav1 channels (but not Cav2 nor Cav3 channels) were demonstrated to have an influence on firing regularity in SNc DA neurons, similar to the one observed for SK channels [16, 100, 99]. Thus, the functional data obtained so far on IH, IA, Ca2+ currents and SK currents (AHP) do not appear consistent with the identified co-localization of channels : co-regulated channels (HCN and Kv4.3) are not part of the same complex [1], and channels belonging to the same complex (Cav1.3 and SK3) do not seem to functionally interact [16, 99]. In a previous study, we demonstrated that several ion channels participating in the control of pacemaking are co-regulated at the genetic level [7]. We postulated that the existence of such a co-expression module ensures that all the ion channels critically involved in pacemaking are expressed altogether, participating in the robustness of the pacemaking process. Similarly, one may argue that the existence of a macromolecular complex comprising Cav1.3, HCN and SK3 channels ensures that all these ion channels will be addressed in the same appropriate compartment (most likely the somatodendritic compartment) and thus participate in the robustness of spontaneous activity.”

Reply to point 2

We added some elements and the corresponding references, on supramolecular complex investigations, in the introduction, line 87.

“Our goal was to identify and quantify interactants of Cav1.2, Cav1.3, HCN2, HCN4, Kv4.3 and SK3 channels using the highly resolutive technique of LC-MS/MS mass spectrometry that has contributed to the robust characterization of many supramolecular signaling complexes encompassing ion channels. Among other studies, Cav2 channels were found to be embedded into protein networks containing around 200 proteins [8], including ion channels, transporters, G protein-coupled receptor-mediated signaling, and release machinery of synaptic vesicles. More recently, the T-type Cav channel interactome [9] was depicted as a “T-type calcium channelosome”.”

Reply to point 3

We discussed the identification and role of ion channel macromolecular complex in the pathophysiology of neurological disorders in the discussion after line 672 in a new chapter called “Putative physiological and physiopathological meaning of the ion channel macromolecular complex”.

“Understanding the role of ion channel complex in the pathophysiology of neurological disease is still in its infancy and lagging behind the comprehension of cardiac dysfunctions. Recently, Maurya et al., [10] identified protein interactants for 13 types of ion channels and established ion channel networks in murine and human heart tissues. Correlations were done in human population genetics data to specifically identify interacting proteins that influence the electrocardiogram [10]. Nevertheless, in neurons, AMPAR interacting partners regulating the number and function of AMPA receptors at the post-synapse, controlling synaptic strength and plasticity have been often found in human patients suffering from schizophrenia or autistic spectrum disorders (see [11] for a review). Additionally, the AMPAR complex-interacting Frrs1l protein was shown to cause severe intellectual disability with cognitive impairment, speech delay and epileptic activity [12]. Furthermore, Niemann-Pick Type C disease was depicted in an animal model as a nanostructural ion channel clustering disease, characterized by alterations in Kv2.1–Cav1.2 nanodomains, which contribute to neurodegeneration [13].”

Finally, in the Results 3.1. section (line 311), Authors should explain better why Kv4.3 proteins " appear more difficult to extract".

We better explained the reading of Figure 1C mentioning that on lane (sol) are revealed channel proteins present in the starting material after solubilization of P2 membrane, on lanes IP1, and IP2 proteins present after immuno-purification in buffers CL47a and CL48, respectively.

Kv4.3 proteins appear more difficult to extract as on Figure 1C, left panel, barely any Kv4.3 channel proteins were revealed on the immunoblot, lane sol. Kv4.3 is mainly seen when immunoprecipitation was done in CL48 buffer. In that case a higher molecular weight band appears that we believed could result of Kv4.3 association with a high molecular weight partially denatured complex pointing to the difficulty to extract Kv4.3 from the membranes.

We did modifications in the Results section, line 309.

“Although HCN2 (Figure 1C, middle panel) and HCN4 (Figure 1C, right panel) channels were observed in the solubilized starting material (sol) before immunoprecipitation, they were not co-purified with Kv4.3 channels immunoprecipitated in buffers CL47a (Figure 1C, lane IP1) or CL48 (Figure 1C, lane IP2), the most powerful buffer for Kv4.3 channel extraction. The same results were obtained in all solubilization buffers of varying stringencies tested (data not shown). However, we cannot exclude any binding of Kv4.3 channels with the identified macromolecular complex as Kv4.3 proteins appear more difficult to extract from the membranes than the other channels (see the absence of Kv4.3 labelling in the solubilized starting material on Figure 1C, left panel, lane sol). Furthermore, Kv4.3-immunolabeled bands of larger apparent molecular weights were observed, indicating possibly the presence of Kv4.3 channels in large partially denaturated complexes.”

REFERENCES

[1]        J. Amendola, A. Woodhouse, M.-F. Martin-Eauclaire, et J.-M. Goaillard, « Ca2+/cAMP-Sensitive Covariation of IA and IH Voltage Dependences Tunes Rebound Firing in Dopaminergic Neurons », J. Neurosci., vol. 32, no 6, p. 2166‑2181, févr. 2012, doi: 10.1523/JNEUROSCI.5297-11.2012.

[2]        C. S. Chan et al., « ‘Rejuvenation’ protects neurons in mouse models of Parkinson’s disease », Nature, vol. 447, no 7148, Art. no 7148, juin 2007, doi: 10.1038/nature05865.

[3]        D. L. Cardozo et B. P. Bean, « Voltage-dependent calcium channels in rat midbrain dopamine neurons: modulation by dopamine and GABAB receptors », J. Neurophysiol., vol. 74, no 3, p. 1137‑1148, sept. 1995, doi: 10.1152/jn.1995.74.3.1137.

[4]        J. Wolfart et J. Roeper, « Selective Coupling of T-Type Calcium Channels to SK Potassium Channels Prevents Intrinsic Bursting in Dopaminergic Midbrain Neurons », J. Neurosci., vol. 22, no 9, p. 3404‑3413, mai 2002, doi: 10.1523/JNEUROSCI.22-09-03404.2002.

[5]        V. de Vrind et al., « Interactions between calcium channels and SK channels in midbrain dopamine neurons and their impact on pacemaker regularity: Contrasting roles of N- and L-type channels », Eur. J. Pharmacol., vol. 788, p. 274‑279, oct. 2016, doi: 10.1016/j.ejphar.2016.06.046.

[6]        M. A. Dufour, A. Woodhouse, et J.-M. Goaillard, « Somatodendritic ion channel expression in substantia nigra pars compacta dopaminergic neurons across postnatal development », J. Neurosci. Res., vol. 92, no 8, p. 981‑999, 2014, doi: 10.1002/jnr.23382.

[7]        M. Tapia et al., « Neurotransmitter identity and electrophysiological phenotype are genetically coupled in midbrain dopaminergic neurons », Sci. Rep., vol. 8, no 1, p. 13637, déc. 2018, doi: 10.1038/s41598-018-31765-z.

[8]        C. S. Müller et al., « Quantitative proteomics of the Cav2 channel nano-environments in the mammalian brain », Proc. Natl. Acad. Sci. U. S. A., vol. 107, no 34, p. 14950‑14957, août 2010, doi: 10.1073/pnas.1005940107.

[9]        N. Weiss et G. W. Zamponi, « The T-type calcium channelosome », Pflugers Arch., vol. 476, no 2, p. 163‑177, févr. 2024, doi: 10.1007/s00424-023-02891-z.

[10]      S. Maurya et al., « Outlining cardiac ion channel protein interactors and their signature in the human electrocardiogram », Nat. Cardiovasc. Res., vol. 2, no 7, Art. no 7, juill. 2023, doi: 10.1038/s44161-023-00294-y.

[11]      D. Bissen, F. Foss, et A. Acker-Palmer, « AMPA receptors and their minions: auxiliary proteins in AMPA receptor trafficking », Cell. Mol. Life Sci., vol. 76, no 11, p. 2133‑2169, 2019, doi: 10.1007/s00018-019-03068-7.

[12]      A. Brechet et al., « AMPA-receptor specific biogenesis complexes control synaptic transmission and intellectual ability », Nat. Commun., vol. 8, p. 15910, juill. 2017, doi: 10.1038/ncomms15910.

[13]      M. Casas et al., « NPC1-dependent alterations in KV2.1–CaV1.2 nanodomains drive neuronal death in models of Niemann-Pick Type C disease », Nat. Commun., vol. 14, no 1, p. 4553, juill. 2023, doi: 10.1038/s41467-023-39937-w.

Reviewer 3 Report

Comments and Suggestions for Authors

The manuscript by Maya Belghazi et al. entitled “High-resolution proteomics unravel a native functional complex of Cav1.3, SK3 and HCN channels in midbrain dopaminergic neurons” describes results from studies on interactions between six ion channels involved in substantia nigra (SNc) neuron autonomous synchronized firing. The topic of the study is particularly interesting since the molecular mechanisms underlying the regulation of the peacemaking activity of these neurons remain poorly understood. Additionally, dopaminergic neurons in SNc are involved in motor control, and their loss is observed in Parkinson’s Disease. I believe that the manuscript deserves publishing in Cells after small corrections:

1.       The results in Figure 1C are unclear and difficult to interpret. The main text and the figure legend should describe these experiments and results more carefully.

2.       Line 307-309: It is mentioned that no molecular interactions were observed using eight different buffers. Are these results ass data not shown? What kind of buffers? Why use eight of them? What is the difference between each buffer?

3.       Line 314: Could the higher migrating band be unspecific?

4.       Line 298 - extra space.

5.       Figure 2 – the legend appears to be duplicated.

6.       Line 376 – how do authors define strong interaction based on MS?

7.       Figure 10 – the legend appears duplicated.

8.       It appears that authors use different fonts in the text (to highlight the most important observations?). I think this should be removed.

Author Response

Review 3-March 19, 2024

Comments and Suggestions for Authors

The manuscript by Maya Belghazi et al. entitled “High-resolution proteomics unravel a native functional complex of Cav1.3, SK3 and HCN channels in midbrain dopaminergic neurons” describes results from studies on interactions between six ion channels involved in substantia nigra (SNc) neuron autonomous synchronized firing. The topic of the study is particularly interesting since the molecular mechanisms underlying the regulation of the peacemaking activity of these neurons remain poorly understood. Additionally, dopaminergic neurons in SNc are involved in motor control, and their loss is observed in Parkinson’s Disease. I believe that       the manuscript deserves publishing in Cells after small corrections:

We thank referee 3 to have helped us to clarify the description and interpretation of our experiments using immunoblots.

  1. The results in Figure 1C are unclear and difficult to interpret. The main text and the figure legend should describe these experiments and results more carefully.

We did modifications in the Results section, line 309.

“Although HCN2 (Figure 1C, middle panel) and HCN4 (Figure 1C, right panel) channels were observed in the solubilized starting material (sol) before immunoprecipitation, they were not co-purified with Kv4.3 channels immunoprecipitated in buffers CL47a (Figure 1C, lane IP1) or CL48 (Figure 1C, lane IP2), the most powerful buffer for Kv4.3 channel extraction. The same results were obtained in all solubilization buffers of varying stringencies tested (data not shown).”

We did modifications in the legend of Figure 1 line 318

“Proteins were membrane-extracted in CL47a solubilization buffer (IP1) or in CL48 buffer (IP2) and immunoprecipitated using anti-Cav1.3 (A), anti-SK3 (B), and anti-Kv4.3 antibodies (C). Blots were performed on starting material: P2 membrane fraction (P2), or on solubilized P2 proteins (sol) or after immunoprecipitation (IP1, IP2). Blots were probed with anti-SK3 (A, right panel), anti-Cav1.3 (B, right panel), anti-HCN4 (A and B left panels; C, right panel), anti-Kv4.3 (C, left panel) and anti HCN2 (C, middle panel). P2: enriched P2 membrane fraction. IP1: immunoprecipitated proteins in CL47a buffer. IP2: immunoprecipitated proteins in CL48 buffer, sol: solubilisate, WB: Western blot (immunoblot).”

  1. Line 307-309: It is mentioned that no molecular interactions were observed using eight different buffers. Are these results ass data not shown? What kind of buffers? Why use eight of them? What is the difference between each buffer?

We performed essay of solubilization using a kit of 8 buffers provided by the company Logopharm providing patented buffers of different compositions and stringencies. The six channels were all solubilized in the CL47a buffer and this buffer was used in all immunoprecipitation experiments. We changed in Figures 1A and 1B “IP” for “IP1” illustrating that immunoprecipitations were done in CL47a buffer as in Figure 1C. We have added the following sentences, after line 297.

“Optimization of extraction and purification conditions were individually searched for the six channel protein complexes testing eight solubilization buffers of varying stringencies from Logopharm (GmbH). The solubilization buffer CL47a able to efficiently extract the six channel subunits from the lipidic membrane was selected to perform co-immunoprecipitation experiments.”

Search for HCN co-purification with Cav1.3, SK3 and Kv4.3 channels were done in eight different buffers. Strong and similar HCN4 signals were obtained in Cav1.3 and SK3 immunoprecipitations made in all eight buffers, strengthening the finding that HCN4 interactions with Cav1.3 and SK3 channels were robust at all different stringencies. Only the signal found in CL47a is shown. We then clarify adding the following sentences line 302

“Based on the literature, we focused on the potential interactions between Cav1.3, Kv4.3, HCN2, HCN4 and SK3 channels. Mouse midbrain plasma membrane-enriched protein fractions were used as starting material (Figure 1A, 1B, lane P2). HCN4 channel subunits were co-immunoprecipitated with Cav1.3 (Figure 1A, left panel, lane IP1) and SK3 (Figure 1B, left panel, lane IP1) channel proteins in buffer CL47a. HCN4 co-purifications were found in all solubilization buffers of varying stringencies tested (data not shown), suggesting strong interactions of HCN4 with Cav1.3 and HCN4 with SK3 channels. Furthermore, SK3 proteins were immunoprecipitated with anti-Cav1.3 antibodies (Figure 1A, right panel) and Cav1.3 subunits with anti-SK3 antibodies (Figure 1B, right panel) in the CL47a solubilization buffer. No molecular interactions were observed between Kv4.3 and HCN2 or HCN4 channels (Figure 1C, middle and right panels, respectively).”

On the contrary when immunoprecipitating Kv4.3 channels in all solubilization buffers, no signal revealing HCN2 and HCN4 was seen. In addition to showing the results using the CL47a buffer we added the one obtained using the CL48 buffer as it was the most powerful to solubilize Kv4.3 channels.The main text has been changed already in that sense, in reply to the point 1, in the Results section, line 309.

“Although HCN2 (Figure 1C, middle panel) and HCN4 (Figure 1C, right panel) channels were observed in the solubilized starting material (sol) before immunoprecipitation, they were not co-purified with Kv4.3 channels immunoprecipitated in buffers CL47a (Figure 1C, lane IP1) or CL48 (Figure 1C, lane IP2), the most powerful buffer for Kv4.3 channel extraction. The same results were obtained in all solubilization buffers of varying stringencies tested (data not shown).”

  1. Line 314: Could the higher migrating band be unspecific?

Although unspecific cross-reactivity of anti-Kv4.3 antibodies cannot be excluded, the higher migrating band on Figure 1C, left panel, was not found using other antibodies like anti HCN2 or anti-HCN4. So, we believe that it is likely that the higher band is Kv4.3 associated with a high molecular weight partially denatured complex. We modified the text line 310

“However, we cannot exclude any binding of Kv4.3 channels with the identified macromolecular complex as Kv4.3 proteins appear more difficult to extract from the membranes than the other channels (see the absence of Kv4.3 labelling in the solubilized starting material on Figure 1C, left panel, lane sol). Furthermore, Kv4.3-immunolabeled bands of larger apparent molecular weights were observed, indicating possibly the presence of Kv4.3 channels in large partially denaturated complexes.”

  1. Line 298 - extra space.
  2. Figure 2 – the legend appears to be duplicated. We modified the sentence to avoid the superimposition of the term “Figure 2”.
  3. Line 376 – how do authors define strong interaction based on MS?

Yes, the term “strong interaction” was not appropriate. We removed it. We meant "significant interactions" line 376

“Unexpectedly, we also found a highly significant interaction of α1-Cav1.2 subunit with Pex5, a protein that associates type 1 peroxisomal targeting signal (PTS1)-containing proteins [38] with peroxisome membranes.”

  1. Figure 10 – the legend appears duplicated. We modified the sentence to avoid the superimposition of the term “Figure 10”.

  1. It appears that authors use different fonts in the text (to highlight the most important observations?). I think this should be removed.

Different fonts were unintentionally created during the PDF edition of the manuscript.

Round 2

Reviewer 1 Report

Comments and Suggestions for Authors

In the revised manuscript, the authors made a number of textual changes (i.e. refined the wording and added information) to address my concerns while adhering to their main conclusions. No additional experiments (particularly IP controls) were performed. Thus, the main issues unfortunately remain unresolved.

To the replies:

- Modification of the conclusion section, line 696: This conclusion is closer to the presented results. However, the term "complex" is still misused given that no evidence is provided on the molecular identity of such an entity. "Biochemical interaction" or "molecular link" may be more adequate terms.

- Specificity of MS/MS analysis: I did not question the quality of the biochemical and mass spectrometric experiments which appear to be overall well conducted. However, I requested additional experiments to rule out antibody-inherent artifacts (like cross reactivities) with target knockout controls and/or additional target-specific antibodies. This is crucial and

independent from the biochemical parameters, replicates or quantification methods used

by the authors to minimize more general non-specific interactors (thoroughly addressed and described in revised paragraph 3.2, line 335).

Mice with full or brain-specific deletion of HCN2, HCN4, SK2, SK3, CaV1.2 or CaV1.3 have in fact been described and would be available as stringent controls.

- PLA assay (manuscript, addition line 492): This is a helpful clarification (yet not a proof for a ternary complex).

- Minor points were mostly resolved as requested. This certainly improved the clarity of the manuscript.

Author Response

Here, find our replies to Reviewer 1:

We thank Reviewer 1 to further improve the value of our manuscript by clarifying the limitations of our results. We did the following changes in the text as requested.

In the revised manuscript, the authors made a number of textual changes (i.e. refined the wording and added information) to address my concerns while adhering to their main conclusions. No additional experiments (particularly IP controls) were performed. Thus, the main issues unfortunately remain unresolved.

To the replies:

- Modification of the conclusion section, line 696: This conclusion is closer to the presented results. However, the term "complex" is still misused given that no evidence is provided on the molecular identity of such an entity. "Biochemical interaction" or "molecular link" may be more adequate terms.

 We did the following modifications:

Line 713:” Putative physiological and physiopathological meaning of the ion channel partnerships”

Lines 716-717: “As summarized in Figure 10, interactions determined here, depict that several ion channels were found to display molecular links in SNc DA neurons: Cav1.3, HCN2, HCN4, and SK3.”

Lines 717-718: “Surprisingly, no molecular interactions with Kv4.3 were observed.”

Lines 738-740: “Thus, the presence of biochemical interactions of Cav1.3 with SK3 contrasts with the demonstrated lack of dependence of SK3 activation on Cav1-mediated calcium entry.”

Lines 751-755: “Similarly, one may argue that the existence of molecular links between Cav1.3, HCN and SK3 channels ensures that all these ion channels will be addressed in the same appropriate compartment (most likely the somatodendritic compartment) and thus participate in the robustness of spontaneous activity.”

Lines 773-776: “Specifically, we identified biochemical interactions between Cav1.3, SK3, HCN2 and HCN4 channels. Ion channel tight proximity would be a molecular basis allowing a fine spatiotemporal control of pacemaking activity.”

Lines 790-795: “Then the molecular links between the hyperpolarization-activated cyclic nucleotide–gated HCN channels, the L-type calcium channel Cav1.3 and the calcium-activated small-conductance SK3 potassium channels identified in midbrain neurons and particularly in SNc DA neurons could help to further understand the underlying factors contributing to the high vulnerability of these neurons and elaborate novel therapies to slow SNc DA neurodegeneration.

- Specificity of MS/MS analysis: I did not question the quality of the biochemical and mass spectrometric experiments which appear to be overall well conducted. However, I requested additional experiments to rule out antibody-inherent artifacts (like cross reactivities) with target knockout controls and/or additional target-specific antibodies. This is crucial and independent from the biochemical parameters, replicates or quantification methods used by the authors to minimize more general non-specific interactors (thoroughly addressed and described in revised paragraph 3.2, line 335). Mice with full or brain-specific deletion of HCN2, HCN4, SK2, SK3, CaV1.2 or CaV1.3 have in fact been described and would be available as stringent controls.

 We added a new paragraph line 566:

 “Limitations and Perspectives

While we've taken great care to minimize any potential unspecific interactions, determining true interactants solely using immunoprecipitation coupled to mass spectrometry is a challenging task. Therefore, further verifications should be conducted to definitively exclude any potential antibody-related artifacts, such as cross-reactivity of certain antibodies. One approach could be to utilize multiple different antibodies targeting the same ion channel and consider only the common interactants identified across these various immunoprecipitations. But undoubtedly, the best option would be to introduce additional negative controls using conditional knock-out mice with substantia nigra dopaminergic neuron deleted of HCN2, HCN4, SK3, Cav1.2, and Cav1.3 at an adult stage to obtain sufficient midbrain membrane material. Such stringent control would serve to corroborate the interactions outlined in this article, thereby enhancing the reliability and validity of our findings.”